# BlockDialect: Block-wise Fine-grained Mixed Format Quantization for Energy-Efficient LLM Inference

**Wonsuk Jang** [1]   **Thierry Tambe** [1]

## Abstract

The rapidly increasing size of large language models (LLMs) presents significant challenges in memory usage and computational costs. Quantizing both weights and activations can address these issues, with hardware-supported fine-grained scaling emerging as a promising solution to mitigate outliers. However, existing methods struggle to capture nuanced block data distributions. We propose BlockDialect, a block-wise fine-grained mixed format technique that assigns a per-block optimal number format from a formatbook for better data representation. Additionally, we introduce DialectFP4, a formatbook of FP4 variants (akin to *dialects*) that adapt to diverse data distributions. To leverage this efficiently, we propose a two-stage approach for online DialectFP4 activation quantization. Importantly, DialectFP4 ensures energy efficiency by selecting representable values as scaled integers compatible with low-precision integer arithmetic. BlockDialect achieves 10.78 % (7.48 %) accuracy gain on the LLaMA3-8B (LLaMA2-7B) model compared to MXFP4 format with lower bit usage per data, while being only 5.45 % (2.69 %) below full precision even when quantizing full-path matrix multiplication. Focusing on *how to represent* over *how to scale*, our work presents a promising path for energy-efficient LLM inference.

## 1. Introduction

Quantization is a crucial technique (Gholami et al., 2022) to address the challenges posed by the exponential growth in Large Language Models (LLMs), including memory bottlenecks (Frantar et al., 2022; Alizadeh et al., 2023; Gho-

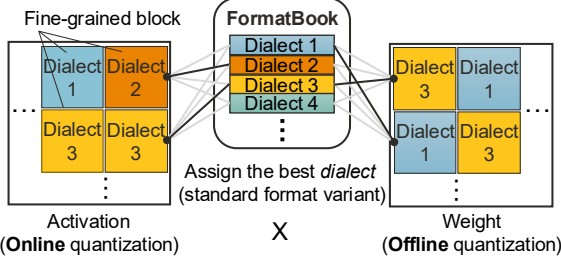

*Figure 1.* Overview of BlockDialect technique workflow.

lami et al., 2024) and increased computational costs (Xiao et al., 2023). By reducing numerical precision, quantization effectively reduces memory usage and data movement overhead (Kim et al., 2023). Additionally, leveraging low-precision operation results in improvements in inference speed, area, and energy efficiency (Xiao et al., 2023; Cao et al., 2024; Rouhani et al., 2023a). A key challenge in LLM quantization lies in handling outliers - elements with much larger magnitudes compared to the rest (Dettmers et al., 2022). Outliers skew the scaling factor, diminishing the representation capacity for the majority of the elements (Liu et al., 2023). To counter this issue, block-wise quantization has been adopted as a common solution (Frantar et al., 2022; Dettmers et al., 2023; Lin et al., 2024; Sheng et al., 2023). By partitioning the tensor into smaller blocks and quantizing each block separately, this method effectively mitigates the influence of outliers within localized areas.

While smaller blocks encapsulate outliers better, they introduce overhead in managing high-precision scaling factors (Rouhani et al., 2023a). To balance this, recent advancements have focused on hardware support for fine-grained block-wise quantization (e.g., block size 32) (Rouhani et al., 2023a; Dai et al., 2021; Rouhani et al., 2023b). In line with this progress, Open Compute Project, backed by leading tech companies, has established the Microscaling (MX) format specification [1]. This format enhances performance and hardware efficiency via fine-grained blocks and power-of-two scaling factors, and has been adopted by recent AI

[1]Department of Electrical Engineering, Stanford University, CA, USA. Correspondence to: Thierry Tambe <ttambe@stanford.edu>.

*Proceedings of the 42nd International Conference on Machine Learning*, Vancouver, Canada. PMLR 267, 2025. Copyright 2025 by the author(s).

---

[1]https://www.opencompute.org/documents/ocp-microscaling-formats-mx-v1-0-spec-final-pdf

accelerators like NVIDIA's Blackwell [2].

In addition to advancements in block-wise quantization, research has progressed to sub-8-bit quantization, reaching even 2-bit precision (Egiazarian et al., 2024; Chee et al., 2024). However, most methods focus on weight-only quantization due to challenges in quantizing activations which are: (1) the need for real-time quantization, (2) a wider dynamic range, and (3) channel-wise variances that misalign with matrix multiplication dimensions (Xiao et al., 2023). As a result, activations remain high-precision, requiring dequantization of weights and high-precision arithmetic operations (Dotzel et al., 2024), which reduces gains in energy efficiency and inference throughput. Moreover, increasing sequence lengths in modern LLMs exacerbate these inefficiencies due to quadratic computational growth (Shyam et al., 2024). Thus, addressing activation quantization is critical for realizing energy-efficient LLM inference.

Our work stems from the insight: "*If a group of numbers deserves its own scaling factor, why not a number format?*" Existing research primarily focuses on "*how to scale*" activations to make them quantization-friendly, often by migrating quantization difficulty to weights (Xiao et al., 2023) or utilizing Hadamard matrices to reduce outliers (Ashkboos et al., 2024). In contrast, we take a novel perspective by exploring "*how to represent*" each block. Leveraging hardware-supported fine-grained block-wise quantization, we propose BlockDialect, which enables 4-bit weight and activation post-training quantization with each block assigned an optimal number format from the formatbook.

Additionally, we present DialectFP4: a formatbook of FP4 variants tailored to diverse block-wise data distributions. To leverage this efficiently, we propose a two-stage approach for online optimal format selection, achieving zero-shot performance comparable to the mean squared error (MSE)-based approach. Importantly, DialectFP4 ensures compatibility with low-precision integer arithmetic by selecting representable values as scaled integers (i.e., multiples of 0.5), enabling the proposed MAC units to achieve the area and energy efficiency of FP4 MAC units. To further enhance energy efficiency, our approach extends to *full-path* matrix multiplication, including not only activation-weight multiplications in linear layers but also activation-activation multiplications in attention blocks.

BlockDialect demonstrates significant improvements over the MXFP4 format, achieving 10.78 % (7.48 %) higher zero-shot accuracy with lower bit usage per data, while showing only 5.45 % (2.69 %) lower accuracy than full precision on the LLaMA3-8B (LLaMA2-7B) for *full-path* quantization. When quantizing only linear layers, BlockDialect achieves a marginal 1.76 % (1.20 %) accuracy drop compared to full

[2]https://www.nvidia.com/en-us/data-center/tensor-cores/

precision. Our contributions can be summarized as follows:

- We introduce BlockDialect, a novel block-wise fine-grained mixed format technique that assigns an optimal number format to each block, enabling accurate representation of data distribution in LLMs.

- We propose DialectFP4, a set of FP4 variants tailored for diverse block-level distributions, and achieve online DialectFP4 activation quantization through a practical two-stage approach, yielding accuracy comparable to an MSE-based approach.

- We demonstrate that our approach outperforms existing methods across multiple LLMs while leveraging low-precision, energy-efficient MAC units.

## 2. Related Work

### 2.1. Block-wise Quantization

Block-wise (or group-wise) quantization is a widely adopted technique that assigns scaling factors on a per-block basis, constraining the impact of outliers within each block. To determine these scaling factors, two methods can be employed: software-supported and hardware-supported. Software-supported methods (Frantar et al., 2022; Dettmers et al., 2023; Lin et al., 2024; Sheng et al., 2023) typically rely on high-precision scaling factors, enhancing accuracy but often require larger blocks due to the overhead of storing and applying scaling factors. In contrast, hardware-supported techniques allow smaller blocks using hardware-friendly scaling factors, such as power-of-two shared exponents. VS-Quant (Dai et al., 2021) and Micro-exponents (Rouhani et al., 2023a) demonstrated the effectiveness of this approach, further enhanced by multi-level scaling factors through dedicated hardware. Open Compute Project recently introduced the microscaling (MX) format (Rouhani et al., 2023b), which uses shared exponents across low-precision formats like FP4 and FP6. Its adoption in recent accelerators [2] highlights ongoing industry efforts to enhance hardware support for fine-grained scaling. Building on these advancements, our work introduces a novel approach that assigns number formats to each fine-grained block.

### 2.2. Non-Uniform Quantization

Non-uniform quantization has been extensively explored as alternatives to integer formats, aiming to better capture data distributions in LLMs. Floating-point formats have proven effective for handling wide value ranges encountered in deep learning models. FP8-Quantization (Kuzmin et al., 2022) highlights how FP8 outperforms INT8 by effectively addressing outliers through its flexible exponent representation. ZeroQuant-FP (Wu et al., 2023) demonstrates that

floating-point formats strike a better balance between dynamic range and precision compared to integer formats.

To improve the flexibility of representable values, lookup-based formats have been studied. NF4 (Dettmers et al., 2024) and SF4 (Dotzel et al., 2024) leverage statistical distribution quantile functions (normal and Student's t distribution, respectively) to better align with LLM profiles. Vector quantization extends this concept by performing vector-level matching with codebooks, with AQLM (Egiazarian et al., 2024) introducing additive codebook quantization and QuIP# (Tseng et al., 2024) proposing cache-efficient compressed codebook. Recently, NxFP (Lo et al., 2024) introduced an enhanced MX format, appending mantissa bits to the shared exponent to address similar observations to ours, such as inaccurate largest magnitude representation (detailed comparison in Appendix D). However, these methods are typically limited to weight-only quantization and depend on high-precision operations, incurring significant compute and energy overhead. Our work uses FP4 variants that capture block-level distributions while ensuring compatibility with low-precision integer arithmetic.

### 2.3. Activation Quantization

Addressing challenges of activation quantization such as real-time execution, large dynamic ranges, and channel-wise outliers, researchers have proposed several approaches: 1) LLM.int8() (Dettmers et al., 2022) and Atom (Zhao et al., 2024) employ mixed precision subgrouping, retaining outliers in high precision. However, this approach incurs non-negligible overhead from handling mixed precision. 2) SmoothQuant (Xiao et al., 2023) migrates the quantization difficulty to weights, enabling low-precision weight and activation quantization, thereby avoiding high-precision operations. 3) Recent advancements using Hadamard matrices reduce outliers while maintaining computational invariance. This enables effective 4-bit weight, activation, and KV cache quantization (Ashkboos et al., 2024; Liu et al., 2024b). However, it incurs overhead from online Hadamard transformation and retains some high-precision components (e.g., queries). 4) Mixed format quantization selects optimal number formats for predefined granularities. For example, LLM-FP4 (Liu et al., 2023) adjusts matrix-wise formats and exponent biases, while MoFQ (Zhang et al., 2024) applies layer-wise format selection between floating point and integer. Yet, they lack adaptability to varying data distributions due to relatively coarse and limited customization strategies.

Our work refines mixed format quantization by introducing FP4 variants with minimal differences in representable values, assigning the optimal variant to fine-grained blocks. To emphasize the use of variants over entirely distinct formats, we use the term ***dialect*** throughout the paper instead of variants.

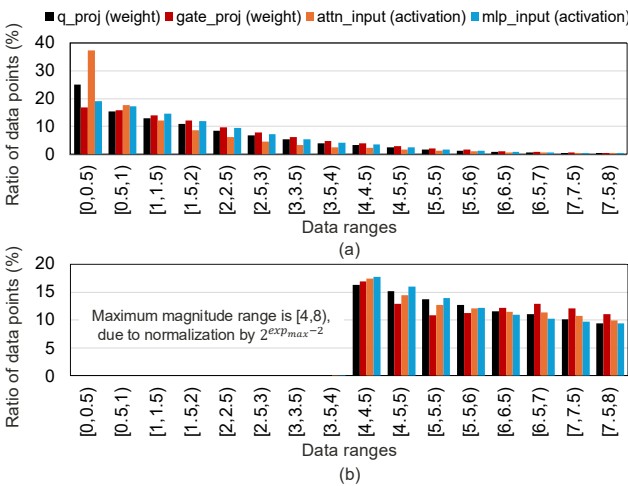

*Figure 2.* LLaMA3-8B block-level profiling results: (a) matrix-wise accumulated magnitude distribution, (b) block's maximum magnitude distribution. Each bar represents the average across layers 0, 10, 20, and 30, with consistent trends across layers.

Additionally, unlike prior methods that rely on calibration or pre-training to reduce online activation processing overhead, BlockDialect supports efficient online processing via two-stage format selection and logical operation-based quantization. It can also capture unstructured outliers through fine-grained block-wise localization, beyond easily calibrated structured outliers (e.g., channel-wise magnitude mean).

## 3. BlockDialect: Block-wise Fine-grained Mixed Format Quantization

To achieve block-wise fine-grained mixed format quantization, we address three key questions: 1) Which *dialects* should be used? 2) How should the per-block *dialect* be selected? 3) How should online quantization and MAC operations be performed?

### 3.1. Which Dialects Should be Used?

**Block-Level Profiling.** To provide a guideline for determining dialects for the formatbook, we conduct profiling of Llama3-8B (Dubey et al., 2024), Llama2-7B (Touvron et al., 2023), Mistral-7B (Jiang et al., 2023), and OPT-6.7B (Zhang et al., 2022) models using WikiText2 (Merity et al., 2016) (Figure 2 shows results for LLaMA3, with results for other models in Appendix A). We split each matrix into blocks of size 32, scale each block by the shared exponent $\lfloor \log_2(\text{block's maximum magnitude}) \rfloor - 2$, and accumulate magnitude distribution histograms for each block. Normalization by the maximum exponent yields values in $[0, 2)$, while FP4 E2M1 spans $[0, 6]$. Subtracting 2 from the shared exponent shifts the range to $[0, 8)$, enabling di-

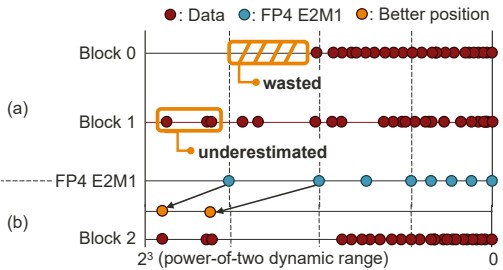

*Figure 3.* Core observations shaping the formatbook design: (a) wasted or underestimated ranges, (b) block-specific distributions deviating from the matrix-wise pattern.

② Different large magnitude distributions

| | | | | | | | | |
|---|---|---|---|---|---|---|---|---|
| Dialect 0 | 7.5 | 5.5 | 3 | 2 | 1.5 | 1 | 0.5 | 0 |
| Dialect 1 | 7.5 | 4.5 | 3 | 2 | 1.5 | 1 | 0.5 | 0 |
| Dialect 2 | 7 | 5.5 | 3 | 2 | 1.5 | 1 | 0.5 | 0 |
| Dialect 3 | 7 | 4.5 | 3 | 2 | 1.5 | 1 | 0.5 | 0 |
| Dialect 4 | 6.5 | 5 | 3 | 2 | 1.5 | 1 | 0.5 | 0 |
| Dialect 5 | 6.5 | 4 | 3 | 2 | 1.5 | 1 | 0.5 | 0 |
| Dialect 6 | 6 | 5 | 3 | 2 | 1.5 | 1 | 0.5 | 0 |
| Dialect 7 | 6 | 4 | 3 | 2 | 1.5 | 1 | 0.5 | 0 |
| Dialect 8 | 5.5 | 4.5 | 3 | 2 | 1.5 | 1 | 0.5 | 0 |
| Dialect 9 | 5.5 | 3.5 | 3 | 2 | 1.5 | 1 | 0.5 | 0 |
| Dialect 10 | 5 | 4.5 | 3 | 2 | 1.5 | 1 | 0.5 | 0 |
| Dialect 11 | 5 | 3.5 | 3 | 2 | 1.5 | 1 | 0.5 | 0 |
| Dialect 12 | 4.5 | 4 | 3 | 2 | 1.5 | 1 | 0.5 | 0 |
| Dialect 13 | 4.5 | 3.5 | 3 | 2 | 1.5 | 1 | 0.5 | 0 |
| Dialect 14 | 4 | 3.5 | 3 | 2 | 1.5 | 1 | 0.5 | 0 |
| Dialect 15 | 4 | 3 | 2.5 | 2 | 1.5 | 1 | 0.5 | 0 |

① Various dynamic ranges    ③ Granularity of 0.5 & common values

*Figure 4.* 16-dialect DialectFP4 example.

rect comparison with FP4 E2M1. Note that we leverage hardware-supported scaling with a power-of-two scaling factor, resulting in a power-of-two dynamic range.

Our matrix-wise analysis revealed that FP4 E2M1 closely aligns with the observed distribution. Specifically, the values are dense in the 0–2 range, sparser between 2–4, and highly sparse between 4–8, patterns that mirror the distribution of representable values in FP4 E2M1 (Figure 2a). Based on this, we select FP4 E2M1 (hereafter referred to as **FP4**) as the base format for our dialects.

However, upon examination of individual block magnitude distributions, we identified two important trends. First, the maximum magnitude of each block is relatively evenly distributed (Figure 2b). Second, some blocks deviate from the overall matrix-wise distribution, which shows more sparsity at the outer bound of the range. For instance, some blocks have multiple values around 7.5 but none in the [4, 7] range. These observations emphasize the importance of aligning with each block's specific distribution, leading to three core principles for our formatbook: 1) minimizing wasted or underestimated ranges, 2) prioritizing the representation of larger magnitudes, and 3) ensuring hardware efficiency.

**Minimizing Wasted or Underestimated Ranges.** Each block has its own dynamic range, but the use of power-of-two shared exponents results in power-of-two dynamic ranges. This mismatch leads to wasted or underestimated range for certain blocks. As illustrated in Figure 3a, blocks with maximum magnitudes smaller than 6 (the maximum representable value of FP4) waste the range beyond their maximum magnitude, as no data points can be mapped to this unused range. For instance, if a block's maximum magnitude is 4.5, the maximum representable value of FP4 (6) could be more effectively utilized by scaling the range to lie between 0 and 4.5. Conversely, data points larger than 6 cannot be represented and are thus underestimated by FP4. This leads to our first criterion: *including FP4 dialects with different maximum magnitudes.*

**Prioritizing the Representation of Larger Magnitudes.**

Large magnitudes, especially outliers, are more likely to yield higher values after multiplication, indicating their greater importance, as similarly noted by other works (Lin et al., 2024; Dettmers et al., 2022). Likewise, we assume that larger magnitudes in each block are also of relatively greater importance. Given the constraints of 4-bit representation, with only 8 distinct representable magnitudes, our approach prioritizes accurately expressing larger magnitudes over smaller ones. Our profiling reveals that each block's scaled data distribution does not always follow the matrix-wise trend, which becomes sparser at the outer bound (Figure 3b). Therefore, simply adjusting the exponent bias is insufficient as it fails to capture the nuanced distributions within specific blocks. Consequently, we establish our second criterion: *generating dialects capable of representing a diverse range of large-magnitude distributions.*

**Ensuring Hardware Efficiency.** To achieve hardware-efficient quantization, our dialects must support low-precision arithmetic. Hence, we maintain a minimum granularity of 0.5. This approach limits the bit width per value and avoids floating-point operations, enabling a more hardware-efficient implementation (detailed in Section 3.3). Also, using multiple dialects requires real-time activation quantization for each selected dialect. This process cannot rely on conventional shift and round logic due to dialect variability, necessitating distinct quantization logic for each. However, implementing separate logic for each value across all dialects would be inefficient. These considerations inform our third criterion: *aligning to 0.5 granularity and preserving most FP4 values across dialects.* This strategy balances representational flexibility with hardware efficiency.

**16-Dialect DialectFP4 Example.** Figure 4 illustrates 16-dialect formatbook, DialectFP4, that meets our three key requirements: ① The dialects cover all possible maximum magnitudes. ② Each pair of dialects shares the maximum

magnitude while differing in one large magnitude value, capturing various large magnitude distributions. ③ The unit of these dialects is 0.5, aligning with FP4, while most of the six smallest magnitude values remain consistent with FP4. Based on this DialectFP4, each data point is stored using 4 bits: 1-bit sign and 3-bit index indicating the value of the selected dialect. Additionally, a 4-bit dialect identifier (for 16 dialects) is assigned to each block.

### 3.2. How Should the Per-Block Dialect be Selected?

While the optimal per-block dialect for preknown weights can be determined by calculating the exact mean square error (MSE) for each dialect, this approach is infeasible for activations due to their dynamic nature. In particular, Block-Dialect focuses on fine-grained activation blocks, where the selection of the dialect has a direct and significant impact on quantization outcomes. This heightened sensitivity demands a more precise and adaptive method. To address this, we adopt a sample dataset-agnostic strategy that performs on-the-fly dialect selection. Specifically, we propose an efficient two-stage selection process that operates after an initial preprocessing step, as shown in Figure 5.

**Preprocessing Stage.** In the preprocessing stage, we compute a 5-bit shared exponent (FP16 exponent bit width) based on the block's maximum magnitude, adjusting it to ensure the expression range, [0,8), fully encompasses FP4's range, [0,6]. Each element's exponent is then adjusted by subtracting the shared exponent, enabling a compact 2-bit exponent per element. For cases where the shared exponent exceeds an element's exponent, a compensatory mantissa shift is applied. The mantissa is then shifted by the adjusted exponent, and the lower bits are truncated to form a 5-bit representation: 3-bit integer part and 2-bit fractional part (Figure 5a). This representation covers DialectFP4's range of 0.0 to 7.5 with 0.5 granularity, requiring 3 integer bits and at least 1 fractional bit. However, to enable accurate rounding during quantization, an additional fractional bit is included, resulting in a 5-bit intermediate format. To clarify, this does not indicate 5-bit quantization; rather, it is an intermediate value used internally during dialect selection and quantization.

**Two-Stage Dialect Selection Process.** Comparing every dialect to find the best dialect for each block is computationally expensive and inefficient. Instead, we adopt a two-stage approach. In the first stage, we narrow down the options by selecting a pair of dialects whose largest magnitudes match the block's maximum (Figure 5b). Recall that each pair of dialects share the maximum magnitude. The block's maximum magnitude can be easily determined by rounding from the second fractional bit ($Block_{maxTrunc}$). This step not only streamlines the selection process but also ensures that the chosen dynamic range aligns with the block's

characteristics, avoiding wasted or underestimated ranges.

In the second stage (Figure 5c), we determine the optimal dialect from the chosen pair by evaluating which one has more block elements within its beneficial range. Since the two dialects differ by only one large magnitude value, the beneficial range is defined as the interval where incorporating this different value reduces quantization error. This range can be calculated as the midpoint between the differing value, its adjacent value, and the paired dialect's differing value. For example, the beneficial range for dialect 4 is [4.5, 5.75), where 4.5 is the midpoint between the differing values of dialect 4 and 5 $((5.0 + 4.0)/2)$, and 5.75 is the midpoint between dialect 4 and its adjacent value $((5.0 + 6.5)/2)$.

A naive approach to count elements within beneficial ranges requires four comparisons (two per each beneficial range) per element, which introduces compute and latency overhead. To optimize this, we convert range checks into efficient logical operations as illustrated in Figure 5c. If we represent each beneficial range as a 5-bit binary representation and enumerate all possible cases, we can compress them into simple logical operations. In Figure 5c, we use $5'b10110$ to represent the upper limit of dialect 4's beneficial range, excluding 5.75 ($5'b10111$). It encompasses all values smaller than 5.75, as we truncate after the second fractional bit. Finally, if the four most significant bits are $4'b1001$, the element falls within dialect 4's beneficial range.

This logic can be pre-designed as we use a fixed DialectFP4. With these simple logical operations, we can efficiently count the number of elements falling within each dialect's beneficial range in parallel. Given that we use fine-grained blocks with sizes up to 64 elements, the counting process can be further optimized using a reduction tree structure.

### 3.3. How Should Online Quantization and MAC Operations be Performed?

DialectFP4 uses a 0.5 granularity for representable values, expressing values from 0 to 7.5 as 4-bit unsigned integers from 0 to 15 $(0.5 * [0-15])$. Consequently, multiplications can be efficiently performed with 4-bit unsigned integer arithmetic, followed by a 2-bit right shift to account for the 0.5 factor of each number. Our quantization target is thus 4-bit (1-bit sign, 3-bit index), dequantized to 5-bit (previous 1-bit sign, 4-bit unsigned integer) before multiplication.

For weights, the optimal dialect for each block is precomputed, with pre-quantization performed prior to inference. During inference, the 3-bit index is converted to 4-bit integers by indexing a pre-stored table of representable values (Figure 6). Most values are shared across dialects, minimizing storage requirements. In contrast, activations require real-time quantization to the nearest representable value of the optimal dialect. For example, consider preprocessed data

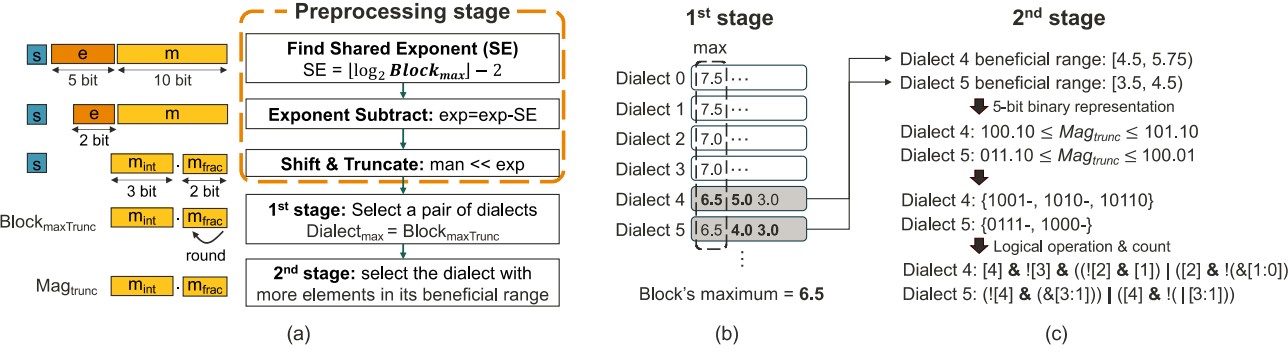

*Figure 5.* Per-block dialect selection process: a) overall process, b) 1st stage, and c) 2nd stage.

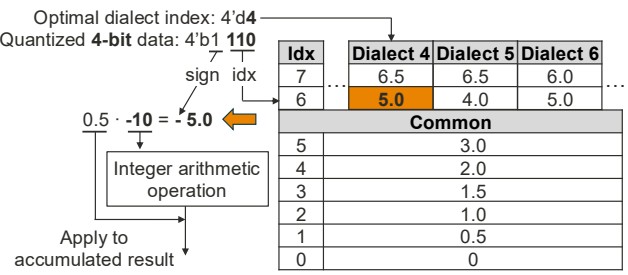

*Figure 6.* Overview of dequantization process.

($Mag_{trunc}$ in Figure 5) as $5'b10001$, representing 8.5 with a 0.5 scaling factor. If the optimal dialect's representable values are $0.5 * [13, 10, 6, 4, 3, 2, 1, 0]$, the value 8.5 would be quantized to 10. Note that this operation cannot be achieved using simple shift and rounding logic. After applying the 0.5 scaling, the final representation is 5.

We can also accelerate this efficiently with a method similar to our dialect selection process. Each representable value is linked to simple checking logic derived from its binary form. For example, data $k$ is quantized to 10 if it falls within [8.0, 11.5), expressed as $[5'b10000, 5'b10110]$. This can be verified by $k[4] \& !k[3] \& !(\&k[2:0])$, which holds true for the previous example, $5'b10001$. The final 4-bit quantized value includes a sign bit and $3'b110$ (the index for 10). The overall quantization logic can be optimized by eliminating redundant logical operations, such as the repeated use of $k[4] \& !k[3]$ across different values. Additionally, since most representable values are shared among dialects, the logic can be further simplified.

While multiplications are performed using 4-bit unsigned integers, all pairwise products between two blocks share the same exponent sum, enabling accumulation with integer operations and resulting in power- and area-efficient hardware (detailed in Section 4.3). Only when all elements in a block are processed, partial sums are converted to FP16

and accumulated, right-shifted by 2 bits to apply the 0.5 scaling factor, and then requantized into 4-bit DialectFP4 format for the next matrix multiplication. Other specialized operations, such as softmax, are performed in high precision. Importantly, the optimal dialect selection, quantization, and MAC operations can be pipelined to further accelerate the overall inference speed.

## 4. Experiments

### 4.1. Experimental Setup

**Models and dataset.** We evaluate BlockDialect on three LLMs: LLaMA-2-7B (Touvron et al., 2023), LLaMA-3-8B (Dubey et al., 2024), and Mistral-7B (Jiang et al., 2023). The evaluation includes seven zero-shot common-sense reasoning tasks: LAMBADA (Paperno et al., 2016), HellaSwag (Zellers et al., 2019), BoolQ (Clark et al., 2019), PIQA (Bisk et al., 2020), WinoGrande (Sakaguchi et al., 2021), ARC-easy, and ARC-challenge (Clark et al., 2018). We leverage the lm-eval-harness (Gao et al., 2023) framework, with *0-shot* notation representing the average accuracy across seven tasks. Additionally, we report perplexity scores on WikiText2 (Merity et al., 2016) with a chunk of 2048.

**Baseline.** We compare BlockDialect with the MXFP4 format (Rouhani et al., 2023b), which employs hardware-supported scaling. Additionally, we compare two recent methods employing software-supported scaling: LLM-FP4 (Liu et al., 2023), and Quarot (Ashkboos et al., 2024). Quarot reduces outliers via Hadamard matrix, while LLM-FP4 adopts a matrix-wise mixed format. We run baselines using their open-source code[3]. To demonstrate BlockDialect's applicability for full-path matrix multiplications, we evaluate two scopes: *linear* (quantizing linear layers) and *all* (extends to attention block operations $QK$, $Attn\_scoreV$).

---
[3] We set (search_interval, search_round) to (60, 2) in LLM-FP4 to avoid excessive calibration time, observing negligible LLaMA-7B accuracy loss compared to the original paper.

*Table 1.* Perplexity on Wikitext2 and average zero-shot accuracy across seven common-sense reasoning tasks. *dn*: down_proj, *Q*: query, *K*: key, *V*: value. †: Quarot keeps query and attention scores in FP16 and performs the associated operations in FP16.

| Scope | Method | Block size (exception) | LLaMA3-8B | | | LLaMA2-7B | | | Mistral-7B | | |
|---|---|---|---|---|---|---|---|---|---|---|---|
| | | | Eff. bit | Wiki↓ | AVG.↑ | Eff. bit | Wiki↓ | AVG.↑ | Eff. bit | Wiki↓ | AVG.↑ |
| - | FP16 | - | 16 | 6.14 | 74.45 | 16 | 5.47 | 70.94 | 16 | 5.32 | 74.92 |
| Linear (*A·W*) | LLM-FP4 | A:tensor, W:ch. | 4 | 48.71 | 41.92 | 4 | 15.61 | 58.15 | 4 | 17.47 | 58.47 |
| | Quarot (W4A4) | A:token, W:ch. | 4 | 8.02 | 66.92 | 4 | 6.04 | 68.00 | 4 | 5.74 | 72.49 |
| | MXFP4 | 16 | 4.31 | 8.20 | 68.53 | 4.31 | 7.07 | 66.86 | 4.31 | 6.49 | 70.33 |
| | | 32 | 4.16 | 8.23 | 68.31 | 4.16 | 7.04 | 65.94 | 4.16 | 6.42 | 70.72 |
| | BlockDialect (w/ DialectFP4) | 32 | 4.28 | **7.05** | 72.24 | 4.28 | **5.84** | **69.74** | 4.28 | **5.65** | **73.46** |
| | | 64 (*dn*:16) | W:4.25 A:4.30 | 7.12 | **72.69** | W:4.23 A:4.27 | 5.88 | 69.51 | W:4.25 A:4.30 | 5.68 | 73.30 |
| | | 64 | 4.14 | 7.30 | 71.51 | 4.14 | 5.96 | 68.95 | 4.14 | 5.75 | 72.76 |
| All (*A·W, A·A*) | Quarot (W4A4KV4) | A:token, W:ch. (*K,V*:128) | W,K,V:4† | 8.17 | 66.01 | W,K,V:4 | **6.10** | 67.50 | W,K,V:4 | **5.80** | 71.72 |
| | MXFP4 | 16 | 4.31 | 18.84 | 58.22 | 4.31 | 11.22 | 60.77 | 4.31 | 9.27 | 66.03 |
| | | 32 | 4.16 | 16.69 | 58.84 | 4.16 | 11.14 | 59.76 | 4.16 | 8.98 | 66.01 |
| | BlockDialect (w/ DialectFP4) | 32 | 4.28 | 7.87 | 68.57 | 4.28 | 6.33 | 67.68 | 4.28 | 5.87 | **72.15** |
| | | 64 (*dn,Q,K*:16) | W:4.25 A:4.21 | **7.77** | **69.00** | W:4.23 A:4.21 | 6.35 | **68.25** | W:4.25 A:4.21 | 5.90 | 71.71 |
| | | 64 | 4.14 | 8.55 | 66.60 | 4.14 | 6.63 | 67.15 | 4.14 | 6.07 | 70.26 |

All operands are quantized along their respective multiplication dimensions, as detailed in Appendix C. We denote *Effective bitwidth (Eff. bit)* as the average bit width required per data, accounting for overhead from scaling factors or dialect identifiers as explained in Appendix I. In the *linear* scope, this metric covers linear layers; in the *all* scope, it includes all matrix multiplications. Block size is 32 unless otherwise specified. For simplicity, we denote each method by its block size, e.g., [method]-32 (block size of 32).

**Implementation.** For performance evaluation, we implement the BlockDialect emulation framework[4] on top of HuggingFace Transformers using PyTorch. All experiments were conducted on a single NVIDIA H100 GPU. For hardware comparison, we model multiply-accumulate (MAC) units for various precision levels using SystemVerilog and synthesize them with Synopsys Design Compiler. The synthesis is performed at 0.5 GHz using the Nangate 45nm OpenCell Library to estimate area and power. Each MAC unit is sized to iteratively add 64 terms from a dot product. For additional prototype hardware cost analysis, we synthesize the design using the SkyWater 130nm standard cell library, targeting a clock frequency of 100 MHz.

### 4.2. Experiment Results

**Main Results.** As shown in Table 1, BlockDialect consistently outperforms MXFP4 across all models in the *linear* scope, even at a lower effective bitwidth. For instance, BlockDialect-64 achieves a 0.93-point lower perplexity and 3.20 % higher accuracy than MXFP4-32 on the LLaMA3 model. Additionally, BlockDialect-64 surpasses both LLM-

---
[4] https://code.stanford.edu/tambe-lab/blockdialect

FP4 and Quarot, with 41.41 / 0.72-point lower perplexity, and 29.59 % / 4.59 % higher accuracy, respectively. Compared to full-precision results, BlockDialect-32 exhibits marginal accuracy drops of 2.21 % / 1.20 % / 1.46 % on the LLaMA3, LLaMA2, and Mistral models, respectively.

BlockDialect achieves full-path quantization (*all*) with minimal performance loss. While MXFP4-16 suffers a significant accuracy drop (~16.23 %), BlockDialect-32 shows resilience, with 5.88 % / 3.26 % / 2.77 % accuracy degradation compared to full precision on the LLaMA3, LLaMA2, and Mistral models, respectively. BlockDialect-32 also outperforms Quarot (W4A4KV4), which quantizes the linear layer, key, and value to 4-bit, demonstrating BlockDialect's superiority. Note that the comparison with W4A4KV4 is conservative, as it retains high-precision components and performs high-precision activation-activation multiplications.

Finally, BlockDialect's performance improves further with smaller blocks in quantization-sensitive sublayers, as detailed in the following block size ablation study. This leads to accuracy drops of only 1.76 % in the LLaMA3 *linear* scope, with 5.45 % / 2.69 % in the *all* scope for the LLaMA3 and LLaMA2 models, respectively, compared to full precision. Additionally, it outperforms MXFP4-16 by 10.78 % / 7.48 % with lower effective bitwidth for these models.

**Comparison with MSE-based Dialect Selection.** We proposed an efficient two-stage approach to eliminate real-time MSE calculation for activation quantization. Table 2 compares our method with the MSE-based approach, where both weight and activation dialects are selected based on MSE. Despite the absence of MSE computations, our approach in *linear* scope results in only a minor perplexity increase (~0.04) and a slight accuracy drop (~0.61 %) across mod-

*Table 2.* Comparison of dialect selection methods.

| Scope | Method | LLaMA3-8B | | LLaMA2-7B | | Mistral-7B | |
|---|---|---|---|---|---|---|---|
| | | Wiki↓ | 0-shot↑ | Wiki↓ | 0-shot↑ | Wiki↓ | 0-shot↑ |
| Linear | MSE | 7.01 | 72.85 | 5.83 | 69.66 | 5.64 | 73.80 |
| | Ours | 7.05 | 72.24 | 5.84 | 69.74 | 5.65 | 73.46 |
| All | MSE | 7.72 | 69.19 | 6.25 | 68.31 | 5.85 | 72.12 |
| | Ours | 7.87 | 68.57 | 6.33 | 67.68 | 5.87 | 72.15 |

*Table 3.* Impact of block size: *dn*: down_proj *o*: output_proj, *q*: q_proj, *k*: k_proj, *v*: v_proj, *Q*: query, *K*: key.

| Scope | Block size (exception) | LLaMA3-8B | | LLaMA2-7B | | Mistral-7B | |
|---|---|---|---|---|---|---|---|
| | | Wiki↓ | 0-shot↑ | Wiki↓ | 0-shot↑ | Wiki↓ | 0-shot↑ |
| Linear | 16 | 6.82 | 72.98 | 5.76 | 69.91 | 5.55 | 73.39 |
| | 32 | 7.05 | 72.24 | 5.84 | 69.74 | 5.65 | 73.46 |
| | 64 | 7.30 | 71.51 | 5.96 | 68.95 | 5.75 | 72.76 |
| | 64 (*dn*:16) | 7.12 | 72.69 | 5.88 | 69.51 | 5.68 | 73.30 |
| | 64 (*o*:16) | 7.24 | 71.68 | 5.94 | 69.60 | 5.73 | 72.64 |
| | 64 (*q,k,v*:16) | 7.19 | 71.66 | 5.91 | 69.28 | 5.68 | 73.30 |
| All | 16 | 7.32 | 70.64 | 6.08 | 68.66 | 5.71 | 72.53 |
| | 32 | 7.87 | 68.57 | 6.33 | 67.68 | 5.87 | 72.15 |
| | 64 | 8.55 | 66.60 | 6.63 | 67.15 | 6.07 | 70.26 |
| | 64 (*dn,Q,K*:16) | 7.77 | 69.00 | 6.35 | 68.25 | 5.90 | 71.71 |

*Table 4.* Comparison across dialect numbers. For the 8-dialect case, two configurations are tested: prioritizing large magnitude distribution (dist.) and dynamic range (max. magnitude) (range).

| Dialect # | LLaMA3-8B | | LLaMA2-7B | | Mistral-7B | |
|---|---|---|---|---|---|---|
| | Wiki↓ | 0-shot↑ | Wiki↓ | 0-shot↑ | Wiki↓ | 0-shot↑ |
| 8 (dist.) | 8.29 | 67.96 | 6.51 | 66.75 | 6.01 | 71.88 |
| 8 (range) | 8.20 | 68.06 | 6.45 | 67.51 | 5.94 | 71.42 |
| 16 | 7.87 | 68.57 | 6.33 | 67.68 | 5.87 | 72.15 |
| 24 | 8.84 | 67.57 | 6.97 | 67.30 | 6.05 | 71.69 |

accurately, leading to reduced performance. The results demonstrate that the 16-dialect formatbook strikes the best balance, effectively addressing both maximum magnitudes and large magnitude distribution. Interestingly, prioritizing maximum magnitude (*range*) results in better performance than prioritizing distribution (*dist.*) overall, aligning with our two-stage approach: select a pair of dialects based on dynamic range first, then choose the better one based on its distribution.

**Additional Exploratory Studies.** To explore various aspects of BlockDialect, we further analyze the dialect selection ratio for each model (Appendix B), confirming that all dialects are meaningful, with no extremely dominant or meaningless dialects. We also evaluate the performance of BlockDialect across various models and workloads (Appendix E, F), demonstrating its generality for LLMs with diverse architectures, sizes, and tasks. In particular, we compare against another baseline format, NVFP4[5], a block scaling format with floating-point scaling factors introduced by NVIDIA. Additionally, we investigate the impact of block shape (Appendix G) and examine the synergy with different activation quantization approaches (Appendix H).

### 4.3. Hardware Cost Analysis

**MAC Unit Comparison.** DialectFP4 is compatible with 5-bit integer arithmetic operations, enabling two implementations: 1) leveraging the general INT4 MAC with simple logic (e.g., shifter) to handle residual bits for 5-bit multiplication (*Ours-INT4*), and 2) designing optimized MACs with 4-bit unsigned integer multiplier and additional XOR gate for sign bit (*Ours*). Although the first option requires more power and area (still efficient than high-precision MACs), it could be a practical option since many commercial accelerators already adopt INT4 MACs. The second option's MAC design achieves area and power efficiency comparable to FP4 (Table 5), providing significantly higher efficiency compared to higher precision MACs. Specifically, it is 1.58 x (1.54 x) more power (area) efficient than FP6 MACs, which are required to achieve better accuracy levels using the MX

els, even when evaluated on the *all* scope (∼0.15, ∼0.63 %), highlighting its effectiveness. This marginal gap also stems from the limitations of the MSE method, which overlooks the magnitude of data elements. This oversight can lead to suboptimal quantization, with inaccuracies for large magnitude values while accurately quantizing smaller ones. By focusing on large magnitudes, our method achieves more efficient and balanced quantization.

**Impact of Block Size.** We explore the impact of block size in Table 3. Overall, smaller sizes improve performance by constraining outliers within smaller blocks and making it easier to represent all block data more effectively with dialects. However, this advantage comes with a higher effective bitwidth, making it a trade-off between performance and memory footprint. We further investigate dynamic block size assignment by applying small blocks to specific projection layers to assess block size sensitivity across sublayers. As in Table 3, down projection has higher sensitivity, which aligns with the observation from prior works (Li et al., 2023; Ashkboos et al., 2023). Based on these findings, we obtain comparable or superior results with block size of 64 by applying smaller blocks only to sublayers prone to outliers (Hooper et al., 2024; Liu et al., 2024a), compared to a uniform block size of 32 in the *all* scope.

**Impact of Number of Dialects.** Table 4 compares the performance across different numbers of dialects in *all* scope. Eight dialects are insufficient to cover both maximum magnitudes and large magnitude distribution, while the 24-dialect formatbook struggles to identify the optimal dialect

---

[5]https://docs.nvidia.com/deeplearning/cudnn/frontend/latest/operations/BlockScaling.html

*Table 5.* Comparison of MAC units with different number formats (0.5GHz, 45nm process). *Ours-INT4* refers to the implementation that leverages the widely adopted INT4 MAC, with supportive logic. Area and power are in $\mu m^2$ and $\mu W$, respectively.

| Type | Multiplier | | Accumulator | | Total | |
|---|---|---|---|---|---|---|
| | Area | Power | Area | Power | Area | Power |
| INT4 | 62.51 | 20.59 | 138.59 | 80.16 | 207.48 | 104.18 |
| INT5 | 101.88 | 34.35 | 171.04 | 106.80 | 275.04 | 142.18 |
| INT8 | 301.91 | 162.93 | 244.72 | 162.79 | 554.34 | 331.17 |
| FP4 | 71.55 | 30.04 | 171.04 | 96.92 | 246.85 | 129.44 |
| FP6 | 158.54 | 73.88 | 223.17 | 139.80 | 381.71 | 213.68 |
| Ours | 63.31 | 16.02 | 184.87 | 118.91 | 248.18 | 134.92 |
| Ours-INT4 | 120.76 | 41.58 | 168.11 | 103.09 | 299.52 | 149.22 |

*Table 6.* Overhead of on-the-fly quantization and dequantization (100MHz, 130nm process).

| Module | Latency (clk cycle) | Power (mW) | Area ($\mu m^2$) |
|---|---|---|---|
| Quantization (incl. format selection) | 5 | 0.7 | 42833.6 |
| Dequantization | 1 | 0.2 | 6319.8 |
| 32 MACs (Ours) | 1 | 2.2 | 41319.6 |
| 32 MACs (INT8) | 1 | 6.1 | 85482.0 |

format (Rouhani et al., 2023b) and 2.45 x (2.23 x) more power (area) efficient than INT8 MACs.

**Overhead of On-the-fly Quantization and Dequantization.** Since the latency and energy benefits - improved computation efficiency through low-precision MACs and reduced data movement via 4-bit quantization - are clear (Yuan et al., 2024; Argerich & Patiño-Martínez, 2024), assessing whether the overhead offsets these gains is crucial. To show this, we synthesize and evaluate an implementation of BlockDialect modules for 32-element processing in SystemVerilog. Since we share six values across dialects, we keep the number of cases manageable, enabling a compact combinational logic implementation. Even with a register file, the overhead remains minimal at 600B for 32-element parallel processing.

As shown in Table 6, quantization and dequantization logic takes only a few clock cycles, which can be further overlapped with pipelining. Their power and area are comparable to or lower than that of our 32 MAC units, indicating minimal overhead. The overhead of on-the-fly activation quantization can also be amortized as the quantized activation block is reused across a large number of weight blocks. Compared to the resources required for INT8 MACs, the practicality of BlockDialect becomes more evident.

**Resource Overhead of Real-time MSE Calculation.** To evaluate the efficiency of our two-stage approach, we compare its resource overhead against that of an MSE-based

*Table 7.* Resource overhead comparison between two format selection methods: two-stage and MSE-based (mean square error).

| Method | Syn. frequency (MHz) | Latency (clk cycle) | Power (mW) | Area ($\mu m^2$) |
|---|---|---|---|---|
| 2-stage | 100 | 5 | 0.7 | 42833.6 |
| MSE | 83.3 | 8 | 6.9 | 399409.3 |

method. Qualitatively, MSE-based method requires 16 rounds (per dialect) of quantization, each involving FP16 square mean error accumulations for every block element, whereas our 2-stage selection efficiently operates in a single pass using 5-bit fixed-point values, logical operations, and simple counting.

Quantitatively, we design in SystemVerilog and synthesize quantization logic implementations with a 130nm process. For a fair comparison, we aim to match the latencies as closely as possible while evaluating area and power. Additionally, since FP16 operations in the exact MSE-based approach incur significant overhead, we convert into fixed-point representation and truncate the lower bits to reduce the complexity. Nevertheless, as shown in table 7, MSE-based logic is 9.32x larger and consumes 9.86x more power. Notably, MSE-based logic fails to meet timing at 100MHz (synthesized at 83.3MHz), whereas our logic meets timing constraints even at 250MHz, underscoring the efficiency of our approach and the impracticality of online MSE-based selection.

## 5. Conclusion

We introduce *BlockDialect*, a post-training quantization technique that assigns an optimal number format to fine-grained blocks. This approach allows the capture of nuanced data distributions often overlooked by existing methods. Complementing this, we develop *DialectFP4*, a set of FP4 variants, which ensure compatibility with an energy- and area-efficient integer MAC unit. To leverage this efficiently, we propose a two-stage approach for online DialectFP4 activation quantization. Our 4-bit quantization results on the LLaMA3-8B (LLaMA2-7B) model show only 5.45 % (2.69 %) accuracy gap compared to full precision for *full-path* quantization. By shifting the focus to how each block should be optimally represented in hardware-efficient manner, rather than solely scaling values, BlockDialect sets a foundation for energy-efficient LLM inference.

## Acknowledgements

We thank Rangharajan Venkatesan (NVIDIA) for his valuable and insightful feedback on our work.

## Impact Statement

This paper introduces advancements in quantization techniques for large language models, focusing on improving memory usage and computational efficiency which, as a societal consequence, could help reduce the significant energy consumption associated with running LLMs, potentially making AI systems more environmentally sustainable and accessible.

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

## A. Block-Level Profiling Results

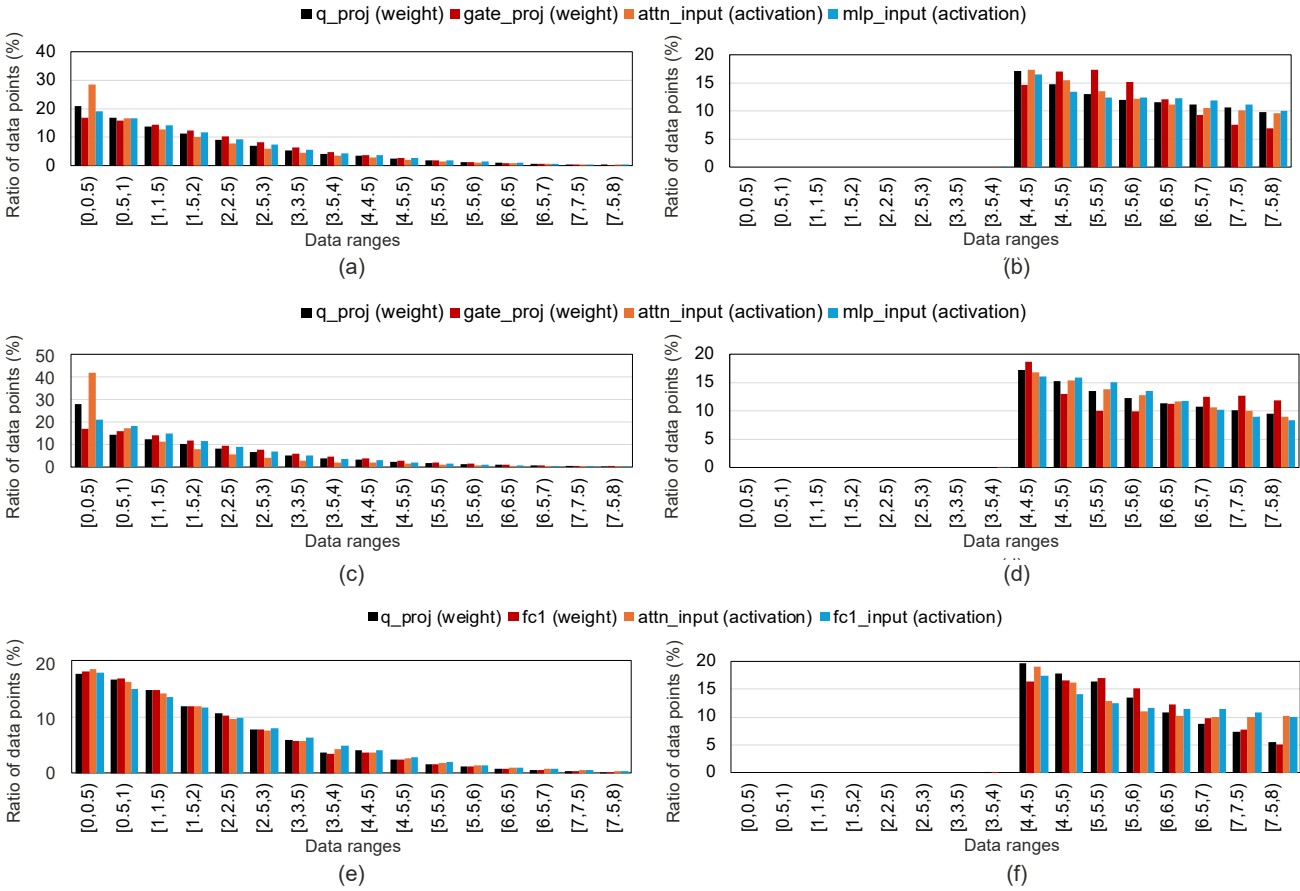

*Figure 7.* LLaMA2-7B (a,b), Mistral-7B (c,d), and OPT-6.7B (e,f) Block-level profiling results: (a), (c), (e) matrix-wise accumulated magnitude distribution, (b), (d), (f) block's maximum magnitude distribution.

Figure 7 presents the block-level profiling results for the LLaMA2-7B, Mistral-7B, and OPT-6.7B models. Each matrix is divided into blocks of size 32, with each block scaled by the shared exponent, $\lfloor \log_2(\text{block's maximum magnitude}) \rfloor - 2$. Magnitude distribution histograms are then accumulated for each block. Figure 7 shows the average results for layers 0, 10, 20, and 30, showing a similar trend to LLaMA3 as discussed in Section 3.1: the matrix-wise accumulated distribution aligns with FP4 E2M1's representable value distribution, while each block's maximum magnitude is relatively evenly distributed across the possible range.

## B. Dialect Selection Ratio

Figure 8 illustrates the selection ratio of each dialect across four models. While the Mistral model shows a slightly higher concentration in selecting specific dialects, all dialects are effectively utilized, with no dialect being overwhelmingly dominant or insignificant. Interestingly, the weights of the Mistral model are more likely to select dialects with larger values (even-number dialects) compared to other models. Unlike weights, activations of all models tend to favor dialects skewed towards smaller values (odd-numbered dialects). It is worth mentioning that a low selection ratio does not necessarily imply low importance for the corresponding dialect. For instance, even if a dialect has a low selection ratio, it can still be valuable if it effectively represents the large magnitudes of certain blocks with high shared exponents, indicating high original magnitudes.

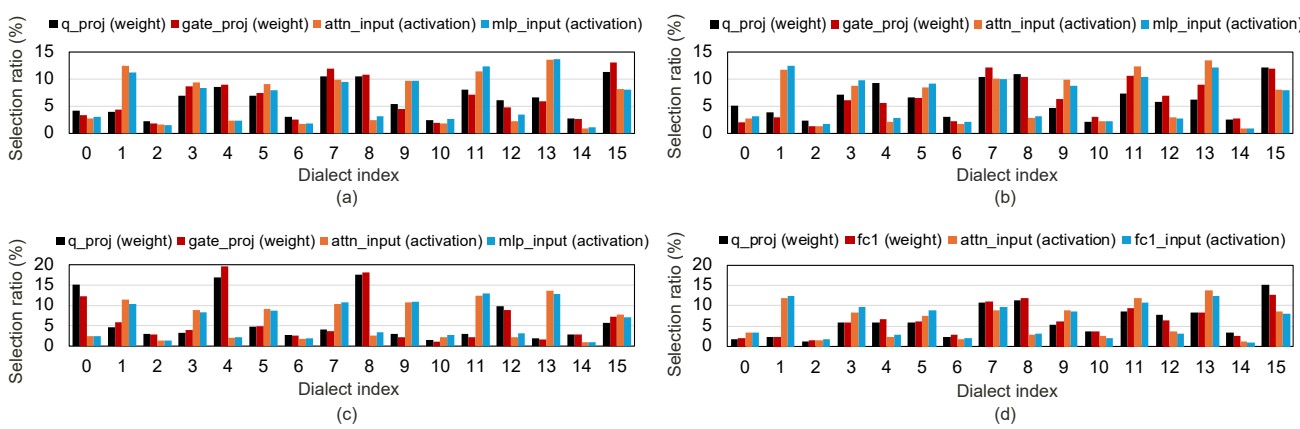

*Figure 8.* Selection ratio of each dialect for (a) LLaMA3-8B, (b) LLaMA2-7B, (c) Mistral-7B, and (d) OPT-6.7B. Experiments were conducted on Wikitext2 with a block size of 32. Each bar represents the average across layers 0, 10, 20, and 30.

## C. Quantization Dimension

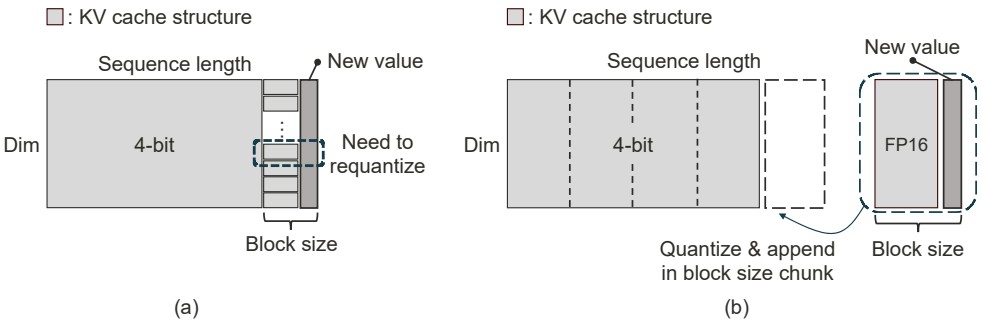

*Figure 9.* Proposed KV cache structure: (a) challenge of sub-channel-wise value quantization, (b) proposed cache structure.

BlockDialect quantizes matrices and vectors along their respective multiplication dimensions. For example, in activation-weight multiplication, activations are quantized at the sub-token level, while weights are quantized at the sub-channel level. Existing KV cache quantization approaches, such as per-token quantization (Sheng et al., 2023), per-channel key quantization with non-uniform representation (Hooper et al., 2024) or group-wise quantization (Ashkboos et al., 2024), primarily focus on compressing and reducing I/O costs during the *decode phase*. These approaches often dequantize data to FP16 before performing multiplications, which limits computational efficiency. In contrast, BlockDialect addresses the full computational path, achieving both memory savings and hardware-efficient computation without FP16 multiplications. Note that BlockDialect's full-path low-precision matrix multiplication is significantly more efficient during the *prefill phase*.

To achieve this, BlockDialect employs sub-token-wise quantization for keys and sub-channel-wise quantization for values, aligning with the respective multiplication dimensions. However, this design introduces a challenge: repeated quantization overhead and numerical inconsistencies when updating the KV cache. Specifically, while sub-token-wise key quantization is straightforward, as new key vectors can be quantized before multiplication and appended to the KV cache, sub-channel-wise value quantization is more complex. When a new value vector is added, the values of each sub-channel must be requantized (Figure 9a). However, BlockDialect discards the FP16 value and directly dequantizes to 5-bit integers with a shared exponent to leverage integer arithmetic operations. As a result, requantizing from the 5-bit integer quantized form risks quantization errors when integrating new value vectors.

To address this, we can leverage *fine-grained* block structure with a default size of 32. Values are stored in 4-bit chunks of *block_size* token count, with only the most recent chunk (size of $N \bmod block\_size$) maintained in high precision. Instead of requantizing all blocks of the most recent chunk for every new value vector, once this chunk's token count reaches the

block size, then the chunk is stored in 4-bit form[6] (Figure 9b). This strategy avoids excessive quantization and ensures accurate updates while keeping the small number of high precision tokens (low portion relative to the total sequence length), resulting in minimal additional storage cost. Similar to Section 4.2, smaller blocks can be used for KV cache quantization, further reducing the overhead and enhancing model performance.

## D. Comparison with NxFP

Concurrent with our work, NxFP (Lo et al., 2024) introduces the Nanoscaling format (NxFP), which improves the limitations of the Microscaling format (MX) (Rouhani et al., 2023b). While NxFP applies to weight-only quantization, requiring high-precision operations, BlockDialect utilizes energy-efficient low-precision integer operations for both weight and activation quantization. However, NxFP and our work originate from the similar observations on the limitations of the MX format and both employ mixed formats, we clearly distinguish our solutions from NxFP's in this section.

NxFP identifies three limitations of the MX format: 1) inaccurate tracking of the largest value, 2) vacant quantization levels inherent to floating point representation, and 3) redundant 0-representations. To address the first issue, NxFP appends a 2-bit mantissa to the shared exponent to improve accuracy of the quantized largest value. The second issue is addressed by selecting the better option between BFP and MxFP. However, these approaches reduce the hardware efficiency of the MX format in scaling factor handling. Also, NxFP proposes techniques similar to other mixed-format quantization strategies, such as choosing between two distinct formats, like integer and floating-point representations (akin to BFP vs. MX after dividing by the shared scaling factor) and adjusting exponent biases to *indirectly* better represent the distribution (akin to additional 2-bit mantissa for shared exponent). While these strategies demonstrate effectiveness, they do not fully account for the fine-grained block-level data distributions that we have observed in our profiling.

In contrast, BlockDialect takes a more tailored approach by selecting multiple (e.g., 16) dialects capable of expressing diverse block-level distributions, while still maintaining a hardware-efficient power-of-two shared exponent. Moreover, BlockDialect dynamically selects dialects that align with block-specific data characteristics, offering a more *direct* and precise approach compared to adjusting exponent biases. Additionally, unlike NxFP, which only proposes MSE-based format selection, we present a practical two-stage approach to enable online activation quantization.

Finally, NxFP and prior works (Dotzel et al., 2024) have proposed remapping redundant 0-representations to other quantization values to improve utilization. While this is also applicable to our work, BlockDialect emphasizes hardware simplicity and minimal dequantization overhead.

---

[6]Similar architecture to the residual cache in https://huggingface.co/blog/kv-cache-quantization, which targets to reduce repeated quantization and preserve accuracy for recent keys and values.

# E. Experimental Results on a OPT-6.7B Model

*Table 8.* Results on **OPT-6.7B** model. Perplexity on Wikitext2 and zero-shot accuracy across seven common-sense reasoning tasks: LAMBADA (LA), WinoGrande (WG), BoolQ (BQ), PIQA (PQ), ARC-easy (A-e), ARC-challenge (A-c), and HellaSwag (HS).

| Scope | Method | Block size | Feature | Eff. bit | Wiki↓ | LA↑ | WG↑ | BQ↑ | PQ↑ | A-e↑ | A-c↑ | HS↑ | AVG.↑ |
|---|---|---|---|---|---|---|---|---|---|---|---|---|---|
| - | FP16 | - | Full precision | 16 | 10.86 | 67.63 | 65.43 | 66.12 | 76.55 | 60.14 | 34.81 | 67.19 | 62.55 |
| Linear | MXFP4 | 16 | HW-supported scaling | 4.31 | 19.17 | 49.78 | 52.41 | 50.06 | 69.31 | 47.39 | 28.92 | 56.64 | 50.64 |
| | | 32 | | 4.16 | 19.22 | 49.89 | 54.93 | 49.54 | 69.10 | 46.89 | 29.44 | 56.04 | 50.83 |
| | BlockDialect (w/ DialectFP4) | 16 | 1D block | 4.56 | 11.26 | 66.89 | 64.48 | 62.57 | 76.55 | 58.71 | 34.90 | 65.20 | 61.33 |
| | | 32 | | 4.28 | 11.31 | 65.73 | 64.40 | 62.11 | 75.68 | 57.58 | 32.17 | 64.69 | 60.34 |
| | | 64 | | 4.14 | 11.73 | 63.21 | 61.33 | 63.43 | 74.81 | 58.12 | 33.02 | 63.16 | 59.58 |
| All | MXFP4 | 16 | HW-supported scaling | 4.31 | 22.94 | 44.93 | 53.67 | 48.65 | 68.34 | 45.45 | 28.33 | 53.99 | 49.05 |
| | | 32 | | 4.16 | 22.12 | 44.11 | 50.43 | 45.54 | 63.49 | 44.15 | 29.35 | 53.81 | 47.27 |
| | BlockDialect (w/ DialectFP4) | 16 | 1D block | 4.56 | 11.45 | 64.87 | 63.54 | 60.70 | 75.19 | 58.71 | 33.11 | 64.71 | 60.12 |
| | | 32 | | 4.28 | 11.63 | 64.62 | 62.19 | 59.66 | 75.41 | 58.29 | 32.85 | 63.82 | 59.55 |
| | | 64 | | 4.14 | 12.14 | 59.42 | 60.85 | 60.43 | 74.10 | 57.28 | 32.34 | 62.86 | 58.18 |

We further evaluate BlockDialect on an LLM with a different architecture, OPT-6.7B (Zhang et al., 2022) in Table 8. Note that the effective bitwidth of BlockDialect-32 (64) is lower than that of MXFP4-16 (32). BlockDialect-32 (64) achieves significant gains over MXFP4-16 (32), showing 7.86 (7.49) and 11.31 (9.98) lower perplexity points, along with 9.70% (8.75%) and 10.50% (10.91%) zero-shot accuracy improvements in *linear* and *all* scopes, respectively. Also, BlockDialect-16 is only 1.22% and 2.43% behind full precision in *linear* and *all* scopes, respectively.

# F. Experimental Results across Architectures, Model Sizes, and Workloads

So far, our evaluation has focused on models with approximately 7-8 billion parameters. To evaluate the general applicability of BlockDialect, we compare performance across a range of architectures, model sizes, and workloads. Specifically, we test LLaMA3-1B, Phi-2.7B (Javaheripi et al., 2023), MobileLLM-125M (Liu et al., 2024c), and GPT2-1.5B (Solaiman et al., 2019). Perplexity (PL) and accuracy of common reasoning (CR) tasks follow the same evaluation setup as in previous experiments. For GLUE (Wang et al., 2018) (GL), we report the average accuracy over six tasks: MRPC, SST-2, RTE, QQP, MNLI, and QNLI. For MMLU (Hendrycks et al., 2020) (ML), we use representative accuracy scores from the lm-eval-harness framework. We evaluate both 16- and 32-block configurations for a comprehensive comparison. Additionally, we include NVFP4 as a baseline - a block scaling format introduced by NVIDIA, which uses FP8 E4M3 floating-point scaling factors and FP4 E2M1 data elements.

Overall, BlockDialect outperforms both data types, demonstrating its versatility, with NVFP4 falling between MXFP4 and

*Table 9.* Performance comparison of BlockDialect (BDFP4), NVFP4, and MXFP4 across various model architectures, sizes, and workloads. *Full* indicates full-path quantization; if unspecified, only linear layers are quantized. MMLU results for GPT2 and MobileLLM are omitted as they are too low to be compared. **Bold** highlights the best result among comparable effective bitwidths (NVFP4-32, MXFP4-16, BDFP4-32).

| Format | Blk size | Eff. bit | LLaMA3-1B | | | | Phi-2.7B | | | | MobileLLM-125M | | | GPT2-1.5B | | | LLaMA3-1B (*Full*) | | | | GPT2-1.5B (*Full*) | | |
|---|---|---|---|---|---|---|---|---|---|---|---|---|---|---|---|---|---|---|---|---|---|---|---|
| | | | PL↓ | CR↑ | ML↑ | GL↑ | PL↓ | CR↑ | ML↑ | GL↑ | PL↓ | CR↑ | GL↑ | PL↓ | CR↑ | GL↑ | PL↓ | CR↑ | ML↑ | GL↑ | PL↓ | CR↑ | GL↑ |
| FP16 | - | - | 9.75 | 60.38 | 37.58 | 52.62 | 9.71 | 72.43 | 54.50 | 64.32 | 12.53 | 46.31 | 51.20 | 17.41 | 53.15 | 48.66 | 9.75 | 60.38 | 37.58 | 52.62 | 17.41 | 53.15 | 48.66 |
| NVFP4 | 16 | 4.5 | 12.40 | 55.18 | 31.96 | 52.72 | 11.28 | 68.76 | 52.09 | 64.88 | 14.91 | 44.18 | 49.63 | 18.60 | 50.88 | 47.65 | 17.46 | 49.16 | 27.15 | 50.81 | 18.81 | 50.42 | 47.76 |
| | 32 | 4.25 | 12.82 | 54.12 | 29.06 | 53.14 | **11.62** | 68.28 | 50.96 | **64.93** | 15.39 | 43.57 | **50.53** | 18.54 | 50.42 | 47.22 | 19.99 | 48.53 | **26.11** | 49.89 | 18.77 | 49.80 | 47.20 |
| MXFP4 | 16 | 4.31 | 15.71 | 51.40 | 27.08 | 50.66 | 12.59 | 69.41 | 50.04 | 61.39 | 18.33 | 42.22 | 49.75 | 19.11 | 51.28 | **48.25** | 53.75 | 41.84 | 24.16 | 49.30 | 20.32 | 49.21 | **47.74** |
| | 32 | 4.16 | 15.91 | 50.60 | 26.26 | 50.55 | 12.83 | 69.09 | 50.01 | 61.09 | 18.16 | 42.18 | 49.24 | 19.00 | 51.28 | 48.03 | 60.04 | 40.04 | 24.05 | 48.91 | 20.07 | 50.00 | 47.53 |
| BDFP4 | 16 | 4.56 | 11.47 | 56.32 | 31.35 | 52.74 | 11.12 | 70.58 | 52.13 | 62.87 | 14.33 | 44.51 | 50.38 | 17.85 | 51.82 | 48.08 | 14.83 | 52.24 | 27.89 | 51.35 | 18.02 | 51.37 | 47.59 |
| | 32 | 4.28 | **12.09** | **55.54** | **30.17** | **53.43** | 11.79 | **69.91** | **52.11** | 64.70 | **15.06** | **43.59** | 50.38 | **18.07** | **51.92** | 47.32 | **17.42** | **49.46** | 25.87 | **51.15** | **18.36** | **51.11** | 46.79 |

BlockDialect. Also, in the full-path results, the accuracy gap widens, solidifying BlockDialect's superiority. Note that unlike BDFP4 and MXFP4, which use a power-of-two shared exponent, NVFP4's floating-point scale factor requires costly floating-point operations (e.g., normalization, scale factor multiplication), with overhead increasing as block size decreases.

## G. Impact of Block Shape

*Table 10.* Impact of block shape: 2D block shapes of sizes 16, 32, and 64 have dimensions of (4,4), A(4,8) or W(8,4), and (8,8), respectively.

| Scope | Block size (shape) | LLaMA3-8B Wiki↓ | 0-shot↑ | LLaMA2-7B Wiki↓ | 0-shot↑ | Mistral-7B Wiki↓ | 0-shot↑ |
|---|---|---|---|---|---|---|---|
| Linear | 16 (1D) | 6.82 | 72.98 | 5.76 | 69.91 | 5.55 | 73.39 |
| | 16 (2D) | 6.88 | 73.16 | 5.82 | 69.43 | 5.58 | 73.31 |
| | 32 (1D) | 7.05 | 72.24 | 5.84 | 69.74 | 5.65 | 73.46 |
| | 32 (2D) | 7.09 | 71.97 | 5.92 | 69.09 | 5.65 | 73.15 |
| | 64 (1D) | 7.30 | 71.51 | 5.96 | 68.95 | 5.75 | 72.76 |
| | 64 (2D) | 7.34 | 71.19 | 6.06 | 69.21 | 5.78 | 72.05 |
| All | 16 (1D) | 7.32 | 70.64 | 6.08 | 68.66 | 5.71 | 72.53 |
| | 16 (2D) | 7.47 | 70.95 | 6.22 | 69.09 | 5.84 | 72.89 |
| | 32 (1D) | 7.87 | 68.57 | 6.33 | 67.68 | 5.87 | 72.15 |
| | 32 (2D) | 7.89 | 68.58 | 6.51 | 67.55 | 5.95 | 72.39 |
| | 64 (1D) | 8.55 | 66.60 | 6.63 | 67.15 | 6.07 | 70.26 |
| | 64 (2D) | 8.50 | 67.57 | 6.87 | 67.47 | 6.13 | 70.75 |

So far, we have experimented exclusively with 1D linear-shaped blocks. However, 2D square-shaped blocks may prove advantageous, as they can better capture channel-wise activation variance compared to sub-token-wise linear-shaped blocks. We compare perplexity and zero-shot common-sense reasoning task accuracy between linear and square-shaped blocks in Table 10. While the 2D block shows slightly better accuracy for *all* scope, there is no clear superiority between 1D and 2D blocks in terms of accuracy. However, 2D block quantization generally results in higher perplexity. We infer that, due to the significant channel-wise variance of the key (Liu et al., 2024a), 2D block quantization for the key in *all* scope results in marginally better accuracy than sub-token-wise 1D block key quantization, while 2D block quantization for the linear layer has minimal impact with small block size.

It is important to note the lm-eval-harness framework processes multiple tokens in parallel, akin to the *prefill phase*. As a result, the reported numbers may not fully capture the impact of block shape during the *decode phase*. In the *decode phase*, operations typically involve GEMV or flat GEMM, which require zero padding for the square shape quantization. This results in an increased effective bitwidth for square-shaped blocks, as the scaling factor is calculated over fewer non-padding elements compared to the full block size. At the same time, it could be beneficial for accuracy, as the effective block size for scaling becomes smaller for padded blocks.

## H. Combination with Other Approach

To explore potential synergy with other approaches, we combine BlockDialect with SmoothQuant (Xiao et al., 2023), which shifts the challenge of activation quantization to the weights. We experiment with various migration strengths ($\alpha$), controlling the aggressiveness of this shift with a granularity of 0.05, and select the most effective one with the lowest perplexity. For a block size of 64, applying SmoothQuant results in 0.09, 0.07, 0.09, and 0.48 points of perplexity improvement, along with 0.03%, 0.13%, 0.26%, and 1.69% accuracy gain in the LLaMA3-8B, LLaMA2-7B, Mistral-7B, and OPT-6.7B models, respectively. This demonstrates an overall improvement from the combination, though the gains are limited in some models.

We hypothesize that, despite the distinct perspectives of BlockDialect and SmoothQuant, they are not entirely orthogonal. Specifically, methods that *flatten* the distribution (like SmoothQuant or other techniques using a rotation matrix) may influence the performance of our approach, which focuses on selecting the best dialect for each distinct *fluctuating* distribution. We believe an optimal balance exists between both approaches. For example, extreme-magnitude outliers could

*Table 11.* Synergistic Effects of combining SmoothQuant with different LLMs.

| Method | Wiki↓ | 0-shot↑ |
|---|---|---|
| LLaMA3-8B w/o SmoothQuant | 7.30 | 71.51 |
| LLaMA3-8B w/ SmoothQuant | 7.21 | 71.54 |
| LLaMA2-7B w/o SmoothQuant | 5.96 | 68.95 |
| LLaMA2-7B w/ SmoothQuant | 5.89 | 69.08 |
| Mistral-7B w/o SmoothQuant | 5.75 | 72.76 |
| Mistral-7B w/ SmoothQuant | 5.66 | 73.02 |
| OPT-6.7B w/o SmoothQuant | 11.73 | 59.58 |
| OPT-6.7B w/ SmoothQuant | 11.25 | 61.27 |

be handled by flattening them using SmoothQuant (or other methods), while moderate outliers could be addressed with BlockDialect. We leave this as an area for future investigation.

## I. Effective Bitwidth Calculation

Effective bitwidth is defined as the average bitwidth required per data element, incorporating overhead from scaling factors and dialect identifiers. Based on FP16 for full precision, a 5-bit shared exponent is used per block for the MX format, contributing an overhead of $5/block\_size$. In BlockDialect, an additional 4-bit overhead per block is required to encode the optimal dialect index (with 16 dialects in the formatbook by default), resulting in a total overhead of $9/block\_size$.

For mixed block sizes, we individually calculate the effective bitwidth for weights and activations to offer a clearer and more precise understanding. The weight calculation is straightforward, but activation quantization considers two possible approaches due to shared activations across multiple projections: 1) weighted summation for shared activations, and 2) counting shared activations only once. The first approach captures computational overhead more accurately, while the second is suited for memory-centric analyses. Since activations are more relevant to computational context, we adopt the first approach. Additionally, for activation-activation multiplications in the attention mechanism, the sequence length affects the effective bitwidth. For example, the attention score involves two dimensions of sequence length, whereas other operands use only one. We base our calculations on a sequence length of 2048. Finally, for per-token or per-channel quantization with software supported high precision scaling factor, we omit overhead calculations as they are negligible.

## J. Full Results

The following tables present the complete experimental results for LLaMA3-8B, LLaMA-7B, and Mistral-7B models.

*Table 12.* Full results on **LLaMA3-8B** model. Perplexity on Wikitext2 and zero-shot accuracy across seven common-sense reasoning tasks: LAMBADA (LA), WinoGrande (WG), BoolQ (BQ), PIQA (PQ), ARC-easy (A-e), ARC-challenge (A-c), and HellaSwag (HS). *dn*: down_proj *o*: output_proj, *q*: q_proj, *k*: k_proj, *v*: v_proj, *Q*: query, *K*: key, and *V*: value. 2D block shapes of sizes 16, 32, and 64 have dimensions of (4,4), A(4,8) or W(8,4), and (8,8), respectively. †: Quarot keeps query and attention scores in FP16 and performs the associated operations in FP16.

| Scope | Method | Block size | Feature | Eff. bit | Wiki↓ | LA↑ | WG↑ | BQ↑ | PQ↑ | A-e↑ | A-c↑ | HS↑ | AVG.↑ |
|---|---|---|---|---|---|---|---|---|---|---|---|---|---|
| - | FP16 | - | Full precision | 16 | 6.14 | 76.05 | 72.77 | 81.38 | 80.79 | 77.74 | 53.24 | 79.17 | 74.45 |
| Linear | LLM-FP4 | A:tensor, W:ch. | Mixed format | 4 | 48.71 | 22.45 | 52.88 | 60.31 | 58.22 | 39.31 | 22.10 | 38.14 | 41.92 |
| | Quarot (W4A4) | A:token, W:ch. | Rotation matrix | 4 | 8.02 | 67.65 | 67.09 | 72.84 | 75.73 | 70.45 | 41.89 | 72.78 | 66.92 |
| | MXFP4 | 16 | HW-supported scaling | 4.31 | 8.20 | 69.18 | 69.77 | 72.91 | 76.93 | 71.21 | 45.99 | 73.72 | 68.53 |
| | | 32 | | 4.16 | 8.23 | 68.14 | 67.01 | 72.66 | 77.15 | 72.73 | 46.93 | 73.57 | 68.31 |
| | | 64 | | 4.08 | 8.34 | 67.03 | 67.09 | 73.06 | 77.09 | 71.63 | 45.56 | 73.27 | 67.82 |
| | BlockDialect (w/ DialectFP4) | 16 | 1D block | 4.56 | 6.82 | 75.10 | 71.74 | 80.76 | 80.41 | 74.75 | 50.77 | 77.35 | 72.98 |
| | | | 2D block | 4.56 | 6.88 | 74.40 | 71.43 | 81.13 | 79.22 | 76.26 | 52.22 | 77.46 | 73.16 |
| | | 32 | 1D block | 4.28 | 7.05 | 73.96 | 72.14 | 78.62 | 78.40 | 74.92 | 50.94 | 76.69 | 72.24 |
| | | | 2D block | 4.28 | 7.09 | 73.80 | 70.96 | 80.15 | 79.33 | 74.83 | 48.12 | 76.57 | 71.97 |
| | | | Exact MSE | 4.28 | 7.01 | 74.09 | 71.03 | 79.57 | 79.92 | 76.60 | 51.71 | 77.02 | 72.85 |
| | | 64 | 1D block | 4.14 | 7.30 | 72.54 | 70.80 | 77.89 | 78.35 | 75.17 | 49.91 | 75.90 | 71.51 |
| | | | 2D block | 4.14 | 7.34 | 73.92 | 70.17 | 77.40 | 78.78 | 73.95 | 48.63 | 76.07 | 71.19 |
| | | | w/ SmoothQuant | 4.14 | 7.21 | 73.04 | 70.24 | 78.17 | 78.89 | 75.13 | 49.57 | 75.74 | 71.54 |
| | | | *dn* block size:16 | W:4.25 A:4.30 | 7.12 | 73.22 | 71.59 | 78.81 | 79.33 | 77.53 | 51.45 | 76.91 | 72.69 |
| | | | *o* block size:16 | W:4.17 A:4.19 | 7.24 | 72.97 | 68.98 | 78.35 | 78.29 | 76.05 | 50.68 | 76.47 | 71.68 |
| | | | *q,k,v* block size:16 | W:4.19 A:4.27 | 7.19 | 73.37 | 70.48 | 77.19 | 79.11 | 75.97 | 49.23 | 76.24 | 71.66 |
| All | Quarot (W4A4KV4) | A:token, W:ch. | *K,V* block size:128 | W,*K,V*:4† | 8.17 | 67.15 | 67.17 | 71.41 | 75.08 | 67.55 | 40.78 | 72.96 | 66.01 |
| | MXFP4 | 16 | HW-supported scaling | 4.31 | 18.84 | 53.00 | 62.51 | 65.14 | 71.55 | 60.73 | 35.58 | 59.01 | 58.22 |
| | | 32 | | 4.16 | 16.69 | 58.49 | 61.25 | 64.74 | 71.22 | 58.04 | 36.18 | 61.99 | 58.84 |
| | BlockDialect (w/ DialectFP4) | 16 | 1D block | 4.56 | 7.32 | 73.24 | 69.69 | 77.71 | 77.69 | 73.11 | 47.01 | 76.06 | 70.64 |
| | | | 2D block | 4.56 | 7.47 | 73.36 | 70.17 | 77.19 | 77.53 | 74.92 | 47.44 | 76.06 | 70.95 |
| | | 32 | 1D block | 4.28 | 7.87 | 71.76 | 66.54 | 74.89 | 76.33 | 71.25 | 44.62 | 74.62 | 68.57 |
| | | | 2D block | 4.28 | 7.89 | 72.04 | 67.01 | 77.13 | 75.68 | 69.23 | 44.03 | 74.96 | 68.58 |
| | | | Exact MSE | 4.28 | 7.72 | 73.03 | 67.17 | 76.27 | 75.68 | 71.30 | 45.99 | 74.91 | 69.19 |
| | | | 8-dialect (dist.) | 4.25 | 8.29 | 70.97 | 66.85 | 74.86 | 75.73 | 69.87 | 44.37 | 73.04 | 67.96 |
| | | | 8-dialect (range) | 4.25 | 8.20 | 70.10 | 66.22 | 75.47 | 75.14 | 70.37 | 44.88 | 74.23 | 68.06 |
| | | | 24-dialect | 4.31 | 8.84 | 70.75 | 66.46 | 74.10 | 75.19 | 68.39 | 44.88 | 73.21 | 67.57 |
| | | 64 | 1D block | 4.14 | 8.55 | 68.02 | 63.54 | 74.13 | 74.92 | 69.32 | 44.54 | 71.70 | 66.60 |
| | | | 2D block | 4.14 | 8.50 | 70.52 | 66.38 | 74.50 | 75.41 | 68.98 | 43.77 | 73.42 | 67.57 |
| | | | *dn,Q,K* block size:16 | W:4.25 A:4.21 | 7.77 | 71.08 | 66.38 | 77.37 | 75.24 | 71.63 | 47.18 | 74.12 | 69.00 |

*Table 13.* Full results on **LLaMA2-7B** model. Perplexity on Wikitext2 and zero-shot accuracy across seven common-sense reasoning tasks: LAMBADA (LA), WinoGrande (WG), BoolQ (BQ), PIQA (PQ), ARC-easy (A-e), ARC-challenge (A-c), and HellaSwag (HS). *dn*: down_proj *o*: output_proj, *q*: q_proj, *k*: k_proj, *v*: v_proj, *Q*: query, *K*: key, and *V*: value. 2D block shapes of sizes 16, 32, and 64 have dimensions of (4,4), A(4,8) or W(8,4), and (8,8), respectively. †: Quarot keeps query and attention scores in FP16 and performs the associated operations in FP16.

| Scope | Method | Block size | Feature | Eff. bit | Wiki↓ | LA↑ | WG↑ | BQ↑ | PQ↑ | A-e↑ | A-c↑ | HS↑ | AVG.↑ |
|---|---|---|---|---|---|---|---|---|---|---|---|---|---|
| - | FP16 | - | Full precision | 16 | 5.47 | 73.88 | 69.06 | 77.77 | 79.05 | 74.58 | 46.25 | 76.00 | 70.94 |
| Linear | LLM-FP4 | A:tensor, W:ch. | Mixed format | 4 | 15.61 | 57.97 | 61.80 | 66.45 | 69.48 | 57.07 | 32.76 | 61.55 | 58.15 |
| | Quarot (W4A4) | A:token, W:ch. | Rotation matrix | 4 | 6.04 | 71.01 | 66.06 | 75.11 | 77.80 | 69.91 | 43.00 | 73.09 | 68.00 |
| | MXFP4 | 16 | HW-supported scaling | 4.31 | 7.07 | 69.84 | 68.11 | 72.14 | 77.31 | 68.35 | 41.38 | 70.86 | 66.86 |
| | | 32 | | 4.16 | 7.04 | 69.73 | 65.51 | 70.89 | 76.61 | 67.59 | 40.36 | 70.91 | 65.94 |
| | | 64 | | 4.08 | 7.05 | 70.58 | 65.11 | 71.25 | 76.50 | 68.35 | 40.70 | 70.81 | 66.19 |
| | BlockDialect (w/ DialectFP4) | 16 | 1D block | 4.56 | 5.76 | 73.92 | 68.67 | 76.54 | 77.64 | 73.74 | 44.03 | 74.82 | 69.91 |
| | | | 2D block | 4.56 | 5.82 | 72.93 | 66.69 | 76.94 | 78.02 | 72.77 | 44.28 | 74.37 | 69.43 |
| | | 32 | 1D block | 4.28 | 5.84 | 73.70 | 69.46 | 76.02 | 78.13 | 72.81 | 43.60 | 74.47 | 69.74 |
| | | | 2D block | 4.28 | 5.92 | 72.27 | 66.30 | 76.85 | 77.80 | 72.73 | 43.43 | 74.24 | 69.09 |
| | | | Exact MSE | 4.28 | 5.83 | 73.24 | 68.27 | 76.88 | 78.13 | 72.39 | 44.28 | 74.44 | 69.66 |
| | | 64 | 1D block | 4.14 | 5.96 | 72.83 | 67.32 | 76.64 | 77.31 | 72.31 | 42.41 | 73.81 | 68.95 |
| | | | 2D block | 4.14 | 6.06 | 72.58 | 67.56 | 77.61 | 77.58 | 71.76 | 43.94 | 73.46 | 69.21 |
| | | | w/ SmoothQuant | 4.14 | 5.89 | 72.27 | 67.17 | 75.87 | 77.48 | 72.64 | 43.60 | 74.54 | 69.08 |
| | | | *dn* block size:16 | W:4.23 A:4.27 | 5.88 | 72.40 | 68.90 | 76.88 | 78.62 | 72.26 | 43.69 | 73.82 | 69.51 |
| | | | *o* block size:16 | W:4.18 A:4.19 | 5.94 | 73.82 | 68.51 | 76.27 | 77.64 | 72.69 | 44.37 | 73.88 | 69.60 |
| | | | *q,k,v* block size:16 | W:4.25 A:4.29 | 5.91 | 73.26 | 67.88 | 76.82 | 77.58 | 72.43 | 42.66 | 74.35 | 69.28 |
| All | Quarot (W4A4KV4) | A:token, W:ch. | *K,V* block size:128 | W,*K*,*V*:4† | 6.10 | 70.79 | 64.33 | 74.40 | 77.20 | 70.12 | 42.92 | 72.72 | 67.50 |
| | MXFP4 | 16 | HW-supported scaling | 4.31 | 11.22 | 60.95 | 61.01 | 66.30 | 74.59 | 61.74 | 35.84 | 64.94 | 60.77 |
| | | 32 | | 4.16 | 11.14 | 60.06 | 60.06 | 65.44 | 73.01 | 59.39 | 35.58 | 64.77 | 59.76 |
| | BlockDialect (w/ DialectFP4) | 16 | 1D block | 4.56 | 6.08 | 72.23 | 64.72 | 76.61 | 76.82 | 71.38 | 44.45 | 74.44 | 68.66 |
| | | | 2D block | 4.56 | 6.22 | 72.06 | 67.40 | 76.27 | 77.48 | 72.52 | 43.34 | 74.55 | 69.09 |
| | | 32 | 1D block | 4.28 | 6.33 | 70.66 | 64.48 | 74.68 | 75.35 | 70.92 | 44.03 | 73.66 | 67.68 |
| | | | 2D block | 4.28 | 6.51 | 70.75 | 67.40 | 73.58 | 76.50 | 69.32 | 41.55 | 73.77 | 67.55 |
| | | | Exact MSE | 4.28 | 6.25 | 71.16 | 66.38 | 75.35 | 77.64 | 71.17 | 42.83 | 73.65 | 68.31 |
| | | | 8-dialect (dist.) | 4.25 | 6.51 | 69.30 | 62.98 | 73.52 | 76.66 | 69.74 | 42.32 | 72.73 | 66.75 |
| | | | 8-dialect (range) | 4.25 | 6.45 | 70.75 | 63.22 | 74.37 | 77.26 | 70.20 | 43.34 | 73.41 | 67.51 |
| | | | 24-dialect | 4.31 | 6.97 | 69.84 | 66.38 | 72.84 | 75.73 | 71.00 | 42.58 | 72.74 | 67.30 |
| | | 64 | 1D block | 4.14 | 6.63 | 70.39 | 65.19 | 73.52 | 75.35 | 69.99 | 43.09 | 72.55 | 67.15 |
| | | | 2D block | 4.14 | 6.87 | 70.60 | 65.67 | 73.91 | 76.71 | 69.57 | 42.92 | 72.94 | 67.47 |
| | | | *dn,Q,K* block size:16 | W:4.23 A:4.21 | 6.35 | 70.81 | 67.17 | 75.44 | 75.84 | 70.96 | 44.28 | 73.28 | 68.25 |

*Table 14.* Full results on **Mistral-7B-v0.3** model. Perplexity on Wikitext2 and zero-shot accuracy across seven common-sense reasoning tasks: LAMBADA (LA), WinoGrande (WG), BoolQ (BQ), PIQA (PQ), ARC-easy (A-e), ARC-challenge (A-c), and HellaSwag (HS). *dn*: down_proj *o*: output_proj, *q*: q_proj, *k*: k_proj, *v*: v_proj, *Q*: query, *K*: key, and *V*: value. 2D block shapes of sizes 16, 32, and 64 have dimensions of (4,4), A(4,8) or W(8,4), and (8,8), respectively. †: Quarot keeps query and attention scores in FP16 and performs the associated operations in FP16.

| Scope | Method | Block size | Feature | Eff. bit | Wiki↓ | LA↑ | WG↑ | BQ↑ | PQ↑ | A-e↑ | A-c↑ | HS↑ | AVG.↑ |
|---|---|---|---|---|---|---|---|---|---|---|---|---|---|
| - | FP16 | - | - | 16 | 5.32 | 75.32 | 73.88 | 82.11 | 82.26 | 78.24 | 52.22 | 80.42 | 74.92 |
| Linear | LLM-FP4 | A:tensor, W:ch. | Mixed format | 4 | 17.47 | 56.92 | 56.27 | 69.24 | 69.64 | 58.42 | 36.26 | 62.55 | 58.47 |
| | Quarot (W4A4) | A:token, W:ch. | Rotation matrix | 4 | 5.74 | 72.75 | 70.24 | 78.87 | 80.58 | 77.44 | 49.57 | 77.99 | 72.49 |
| | MXFP4 | 16 | HW-supported scaling | 4.31 | 6.49 | 71.43 | 68.43 | 75.38 | 79.33 | 74.24 | 47.27 | 76.24 | 70.33 |
| | | 32 | | 4.16 | 6.42 | 71.80 | 70.80 | 75.47 | 79.71 | 74.54 | 46.59 | 76.16 | 70.72 |
| | | 64 | | 4.08 | 6.46 | 70.89 | 67.96 | 75.11 | 79.27 | 74.37 | 46.67 | 75.95 | 70.03 |
| | BlockDialect (w/ DialectFP4) | 16 | 1D block | 4.56 | 5.55 | 73.80 | 70.56 | 80.43 | 81.45 | 77.36 | 50.60 | 79.54 | 73.39 |
| | | | 2D block | 4.56 | 5.58 | 72.68 | 70.64 | 80.86 | 81.39 | 77.36 | 50.77 | 79.49 | 73.31 |
| | | 32 | 1D block | 4.28 | 5.65 | 73.45 | 71.11 | 81.13 | 81.83 | 77.31 | 50.17 | 79.20 | 73.46 |
| | | | 2D block | 4.28 | 5.65 | 73.41 | 70.64 | 80.00 | 80.90 | 77.65 | 50.43 | 79.00 | 73.15 |
| | | | Exact MSE | 4.28 | 5.64 | 74.40 | 71.35 | 81.38 | 80.85 | 77.99 | 51.28 | 79.38 | 73.80 |
| | | 64 | 1D block | 4.14 | 5.75 | 73.08 | 69.85 | 79.63 | 80.69 | 77.19 | 49.83 | 79.04 | 72.76 |
| | | | 2D block | 4.14 | 5.78 | 71.18 | 66.69 | 80.64 | 80.69 | 76.52 | 50.68 | 77.95 | 72.05 |
| | | | w/ SmoothQuant | 4.14 | 5.66 | 73.06 | 71.59 | 80.98 | 80.09 | 77.06 | 49.49 | 78.84 | 73.02 |
| | | | *dn* block size:16 | W:4.25 A:4.30 | 5.68 | 73.20 | 70.17 | 80.28 | 81.34 | 77.74 | 51.02 | 79.37 | 73.30 |
| | | | *o* block size:16 | W:4.17 A:4.19 | 5.73 | 72.79 | 69.69 | 78.38 | 80.52 | 77.40 | 50.43 | 79.27 | 72.64 |
| | | | *q,k,v* block size:16 | W:4.19 A:4.27 | 5.68 | 73.36 | 72.14 | 79.51 | 80.69 | 77.95 | 50.34 | 79.11 | 73.30 |
| All | Quarot (W4A4KV4) | A:token, W:ch. | *K,V* block size:128 | W,*K*,*V*:4† | 5.80 | 73.10 | 68.35 | 79.30 | 79.16 | 76.94 | 47.35 | 77.81 | 71.72 |
| | MXFP4 | 16 | HW-supported scaling | 4.31 | 9.27 | 64.64 | 64.40 | 70.89 | 77.15 | 69.99 | 42.92 | 72.23 | 66.03 |
| | | 32 | | 4.16 | 8.98 | 63.90 | 65.90 | 71.41 | 76.39 | 69.57 | 43.09 | 71.81 | 66.01 |
| | BlockDialect (w/ DialectFP4) | 16 | 1D block | 4.56 | 5.71 | 72.50 | 69.69 | 79.60 | 80.36 | 76.56 | 49.91 | 79.07 | 72.53 |
| | | | 2D block | 4.56 | 5.84 | 72.81 | 69.38 | 80.46 | 80.90 | 76.85 | 51.11 | 78.74 | 72.89 |
| | | 32 | 1D block | 4.28 | 5.87 | 71.69 | 69.85 | 80.28 | 80.36 | 75.93 | 48.55 | 78.37 | 72.15 |
| | | | 2D block | 4.28 | 5.95 | 71.86 | 69.30 | 79.54 | 81.28 | 77.57 | 48.98 | 78.23 | 72.39 |
| | | | Exact MSE | 4.28 | 5.85 | 71.63 | 69.06 | 80.40 | 80.41 | 75.72 | 49.06 | 78.57 | 72.12 |
| | | | 8-dialect (dist.) | 4.25 | 6.01 | 72.06 | 68.82 | 81.22 | 80.03 | 75.25 | 48.12 | 77.64 | 71.88 |
| | | | 8-dialect (range) | 4.25 | 5.94 | 71.51 | 68.19 | 79.69 | 79.87 | 75.67 | 46.93 | 78.10 | 71.42 |
| | | | 24-dialect | 4.31 | 6.05 | 71.01 | 69.38 | 79.14 | 80.58 | 75.04 | 48.89 | 77.82 | 71.69 |
| | | 64 | 1D block | 4.14 | 6.07 | 70.00 | 67.48 | 76.73 | 79.22 | 74.45 | 46.84 | 77.08 | 70.26 |
| | | | 2D block | 4.14 | 6.13 | 69.49 | 66.30 | 78.84 | 79.71 | 75.29 | 48.63 | 77.02 | 70.75 |
| | | | *dn,Q,K* block size:16 | W:4.25 A:4.21 | 5.90 | 71.20 | 69.93 | 78.44 | 79.43 | 75.55 | 49.49 | 77.96 | 71.71 |

