# OpenReview forum: "BlockDialect: Block-wise Fine-grained Mixed Format Quantization for Energy-Efficient LLM Inference"
_ICML.cc/2025/Conference — ICML 2025 poster_

### Official Review · Reviewer_8c5N · 2025-02-20

**Overall Recommendation:** 3

**Summary:**

This paper introduces BlockDialect, a block-wise fine-grained mixed format quantization technique designed to enhance the energy efficiency of large language model (LLM) inference. Unlike traditional quantization methods that focus on scaling values, BlockDialect assigns a number format to each block using a predefined formatbook, DialectFP4, which consists of multiple FP4 variants (dialects) tailored to different data distributions. The proposed two-stage selection process aims to determine the best dialect for each activation block in real-time, ensuring compatibility with low-precision integer arithmetic. Experimental results demonstrate that BlockDialect outperforms MXFP4 format, achieving up to a 10.78% accuracy gain on LLaMA3-8B while maintaining lower bit usage and only a 5.45% drop compared to full precision.

**Claims And Evidence:**

Please refer to **“Methods And Evaluation Criteria”**.

**Essential References Not Discussed:**

N/A.

**Experimental Designs Or Analyses:**

Please refer to **“Methods And Evaluation Criteria”**.

**Methods And Evaluation Criteria:**

The accuracy of the proposed method BlockDialect (DialectFP4) has been verified across different models and evaluation tasks, such as perplexity (PPL) and zero-shot downstream tasks. The evaluation results demonstrate BlockDialect can preserve the model’s accuracy better than baselines such as MXFP4.

However, one of the important baselines, NVFP4, is missing in the evaluation. As [NVFP4](https://docs.nvidia.com/deeplearning/cudnn/frontend/latest/operations/BlockScaling.html) supports floating-point micro-scaling factors, it can preserve the model’s accuracy better than MXFP4.

For the efficiency benchmarks, the paper mainly presents the area and energy consumption of MAC units, while the costs (latency, memory, and energy) of on-the-fly quantization & dequantization units are missing.

**Other Comments Or Suggestions:**

N/A.

**Other Strengths And Weaknesses:**

Please refer to **“Methods And Evaluation Criteria”**.

**Questions For Authors:**

1) How is the accuracy of BlockDialect compared to NVFP4 quantization?
2) What is the overhead for the 2-stage block-wise quantization in BlockDialect?
3) Is the per-block dialect design compatible with prevalent GEMM accelerator architectures such as tensor core?

**Relation To Broader Scientific Literature:**

The paper contributes to accelerators for quantized LLM inference, with a particular focus on improving FP4 data format’s accuracy.

**Theoretical Claims:**

N/A.

---

> ### Author Rebuttal · Authors · 2025-03-31
>
> **Performance comparison of BlockDialect (BDFP4), NVFP4, and MXFP4 across various model sizes and architectures**
> * We compare accuracy and perplexity for both `Linear` layer quantization and `Full`-path (including activation-activation multiplication) quantization. Perplexity (PL) and common reasoning tasks (CR) evaluations follow the BlockDialect paper; for GLUE (GL), we average six tasks (MRPC, SST2, RTE, QQP, MNLI, QNLI), and for MMLU (ML), we use the representative accuracy from lm-eval-harness framework. We test both 16- and 32-block sizes for a comprehensive comparison.
> * Overall, BlockDialect outperforms both data types, demonstrating its versatility, with NVFP4 falling between MXFP4 and BlockDialect. Also, in the full-path results, the accuracy gap widens, solidifying BlockDialect’s superiority.
> * Note that unlike BDFP4 and MXFP4, which use a power-of-two shared exponent, NVFP4’s floating-point scale factor requires costly floating-point operations (e.g., normalization, scale factor multiplication), with overhead increasing as block size decreases.
>
> ###### N/A: Too low to compare
> ###### **Bold**: Best result among comparable effective bitwidths (NVFP4-32, MXFP4-16, BDFP4-32)
> ||`Linear`||LLaMA3-1B|OPT-6.7B|Phi-2.7B|MobileLLM-125M|`Full`LLaMA3-1B|
> |:-:|:-:|:-:|:-:|:-:|:-:|:-:|:-:|
> |**Format**|**BlkSize**|**Eff.bit.**|PL↓\|CR↑\|ML↑\|GL↑|PL↓\|CR↑\|ML↑\|GL↑|PL↓\|CR↑\|ML↑\|GL↑|PL↓\|CR↑\|ML↑\|GL↑|PL↓\|CR↑\|ML↑\|GL↑
> |FP16|-||9.8 \|60.4\|37.6\|52.6|10.9\|62.6\|N/A\|52.8|9.7\|72.4\|54.5\|64.3|12.5\|46.3\|N/A\|51.2|9.8\|60.4\|37.6\|52.6
> |NVFP4|16|4.5|12.4\|55.2\|32.0\|52.7|12.4\|59.3\|N/A\|47.9|11.3\|68.8\|52.1\|64.9|14.9\|44.2\|N/A\|49.6|17.5\|49.2\|27.2\|50.8
> ||32|4.25|12.8\|54.1\|29.1\|53.1|12.5\|59.2\|N/A\|**51.6**|**11.6**\|68.3\|51.0\|**64.9**|15.4\|43.6\|N/A\|**50.5**|20.0\|48.5\|**26.1**\|49.9
> |MXFP4|16|4.31|15.7\|51.4\|27.1\|50.7|19.2\|50.6\|N/A\|49.9|12.6\|69.4\|50.0\|61.4|18.3\|42.2\|N/A\|49.8|53.8\|41.8\|24.2\|49.3
> ||32|4.16|15.9\|50.6\|26.3\|50.6|19.2\|50.8\|N/A\|50.3|12.8\|69.1\|50.0\|61.1|18.2\|42.2\|N/A\|49.2|60.0\|40.0\|24.1\|48.9
> |BDFP4|16|4.56|11.5\|56.3\|31.4\|52.7|11.3\|61.3\|N/A\|51.3|11.1\|70.6\|52.1\|62.9|14.3\|44.5\|N/A\|50.4|14.8\|52.2\|27.9\|51.4
> ||32|4.28|**12.1**\|**55.5**\|**30.2**\|**53.4**|**11.3**\|**60.3**\|N/A\|51.3|11.8\|**69.9**\|**52.1**\|64.7|**15.1**\|**43.6**\|N/A\|50.4|**17.4**\|**49.5**\|25.9\|**51.2**
>
> **Overhead for the 2-stage format selection**
> * We evaluate the overhead of the 2-stage format selection to highlight its superiority over the conventional Mean Square Error method. Please see our response to Reviewer 2 (ID: 6poj) in “Resource overhead of real-time MSE calculation” section.
>
> **Compatibility with prevalent GEMM accelerator architectures**
> * Given that BlockDialect uses 4-bit unsigned integer MAC and a logical operation-based quantization method, it can be integrated into existing GEMM accelerators. Moreover, as block-wise operations become more common and commercial accelerators start supporting block-wise quantization formats, integrating BlockDialect will become increasingly seamless.
>
> **Overhead of on-the-fly quantization/dequantization**
> * We synthesize and evaluate an implementation of BlockDialect modules for 32-element processing in SystemVerilog, as an alternative to implementing the GPU kernel within the time constraints. Since we share six values across dialects, we keep the number of cases manageable, enabling a compact combinational logic implementation. Even with a register file, the overhead remains minimal at 600B for 32-element parallel processing. The reported numbers use the 130nm process node at 100MHz.
>
> * Since the latency and energy benefits - improved computation efficiency through low-precision MACs and reduced data movement via 4-bit quantization - are clear$^{[1,2]}$, assessing whether the overhead offsets these gains is crucial. As shown in the results, quantization and dequantization logic takes only a few clock cycles, which can be further overlapped with pipelining. Their power and area are comparable to or lower than that of our 32 MAC units, indicating minimal overhead. The overhead of on-the-fly activation quantization can also be amortized as the quantized activation block is reused across a large number of weight blocks. Compared to the resources required for INT8 MACs, the practicality of BlockDialect becomes more evident. To fully realize BlockDialect’s potential, we are taping out an optimized accelerator that will provide end-to-end measurements.
>   - ###### [1] Yuan et al., "LLM Inference Unveiled: Survey and Roofline Model Insights," arXiv, 2024. [2] Argerich & Patiño-Martínez, "Measuring and improving the energy efficiency of large language models inference," IEEE Access, 2024.
>
>
>
> |Module|Latency ($clock\ cycle$)|Power ($mW$)|Area ($\mu m^2$)|
> |-|:-:|:-:|:-:|
> |Quantization (including format selection)|5|0.7|42833.6|
> |Dequantization|1|0.2|6319.8|
> |32 MACs (Ours)|1|2.2|41319.6|
> |32 MACs (INT8)|1|6.1|85482.0|

---

### Official Review · Reviewer_rDoa · 2025-03-05

**Overall Recommendation:** 3

**Summary:**

This work presents BlockDialect, a block-wise mixed format quantization method for energy-efficient LLM inference. It assigns each block an optimal number format from a predefined formatbook to better capture data distributions. The proposed DialectFP4, a set of FP4 variants, enhances flexibility while maintaining hardware efficiency. A two-stage online format selection method enables efficient activation quantization without costly MSE calculations.

**Claims And Evidence:**

Most of the claims in the paper are supported by strong empirical evidence and theoretical analysis, particularly regarding accuracy improvements, hardware efficiency, and training stability. However, while the method is evaluated on LLaMA3-8B, LLaMA2-7B, and Mistral-7B, it remains unclear how well BlockDialect generalizes to other architectures such as GPT, OPT, or hybrid transformer-based models.
Additional experiments on more diverse model families would strengthen this claim. Also, Not entirely. The method claims to be energy-efficient, but the evaluation of energy consumption provided in Table 5 seems not support this claim.

**Essential References Not Discussed:**

N/A

**Experimental Designs Or Analyses:**

Yes, I have reviewed the soundness and validity of the experimental designs and analyses presented in the submission. Overall, the experimental setup is reasonable.

However, there are still some points that can be improve:
- While the paper claims that BlockDialect is energy-efficient, it does not provide concrete measurements of energy savings in actual LLM inference workloads. The inclusion of MAC unit power analysis is useful but does not fully capture real-world energy consumption across the entire inference pipeline. A more rigorous validation of this claim would involve reporting end-to-end energy usage when running inference on real hardware accelerators, such as GPUs or TPUs.

- The experiments focus primarily on LLaMA and Mistral models, leaving uncertainty about whether BlockDialect generalizes to other architectures, such as GPT, OPT, or hybrid transformer-based models. Given that different model families may exhibit varying sensitivities to quantization, additional evaluations on a broader range of LLM architectures would provide a more comprehensive validation of the approach.

- Since BlockDialect assigns per-block format identifiers, there is an inherent trade-off between format flexibility and additional metadata storage requirements. However, the paper does not quantify how much extra memory is needed to store these identifiers or discuss scalability concerns for very large models. Providing a detailed breakdown of the memory overhead and its impact on inference efficiency would enhance the clarity of this trade-off.

- While the hardware synthesis results demonstrate energy and area efficiency at the MAC unit level, there is no empirical validation of how BlockDialect affects overall inference latency when deployed on real hardware. Without these benchmarks, it remains unclear whether the proposed method actually accelerates practical LLM inference. Including end-to-end latency measurements would significantly strengthen the practical relevance of BlockDialect.

**Methods And Evaluation Criteria:**

Yes, the proposed methods and evaluation criteria are well-structured and mostly relevant to the quantization challenges in LLM inference.

**Other Comments Or Suggestions:**

See Other Strengths And Weaknesses

**Other Strengths And Weaknesses:**

One of the key strengths of the paper is its practical approach to energy-efficient quantization by introducing a block-wise mixed-format strategy that balances flexibility and efficiency.

However, I still have questions about the novelty of the paper, as it builds upon existing mixed-precision and block-wise quantization techniques rather than introducing fundamentally new principles. While the format book selection and DialectFP4 variants add flexibility, the core idea of assigning optimal numerical formats based on block-wise distributions has been explored in various forms before such as [1-3].

[1]Liu R, Wei C, Yang Y, et al. Block-wise dynamic-precision neural network training acceleration via online quantization sensitivity analytics[C]//Proceedings of the 28th Asia and South Pacific Design Automation Conference. 2023: 372-377.
[2]Wu X, Hanson E, Wang N, et al. Block-Wise Mixed-Precision Quantization: Enabling High Efficiency for Practical ReRAM-based DNN Accelerators[J]. IEEE Transactions on Computer-Aided Design of Integrated Circuits and Systems, 2024.
[3]Dettmers T, Lewis M, Shleifer S, et al. 8-bit optimizers via block-wise quantization, 2022[J]. URL https://arxiv. org/abs/2110.02861, 2022.

**Questions For Authors:**

See above.

**Relation To Broader Scientific Literature:**

This work builds on prior research in block-wise and mixed-precision quantization, extending methods like MXFP4 and LLM-FP4 by introducing fine-grained format selection with DialectFP4. It aligns with recent advancements in adaptive numerical precision and hardware-efficient LLM training, particularly in optimizing low-power MAC operations.

**Theoretical Claims:**

N/A

---

> ### Author Rebuttal · Authors · 2025-03-31
>
> **Evaluating the impact of BlockDialect on inference latency and energy consumption**
> * That’s a valid point. Given the clear energy and latency benefits from low-precision MACs and reduced data movement due to 4-bit weight/activation quantization (including KV cache), we assess the overhead from BlockDialect's additional logic across various aspects to determine if these benefits can be offset. We address this in our response to Reviewer 4 (ID: 8c5N) (see the "Overhead of on-the-fly quantization/dequantization" section).
>
> **Broader applicability of BlockDialect to architectures other than LLaMA and Mistral**
> * In appendix E, we provide experiment results for the OPT-6.7B model. Also, you can refer to our response to Reviewer 4 (ID: 8c5N), section "Performance comparison of BlockDialect, NVFP4, and MXFP4 across various model sizes and architectures," for further experiments including OPT and Phi models.
>
> **Metadata storage requirements**
> * As mentioned on page 6, line 306, "Effective bitwidth" indicates the per-block metadata overhead and we provide the detailed calculation method in Appendix H (page 16). Our results show that BlockDialect achieves better accuracy than the MX format, even with a lower effective bitwidth. Additionally, metadata overhead is mitigated by sharing it across the block, reducing it to less than 0.3 bits per data. Note that even with larger models, the per-block data and metadata ratio remains constant.
>
> **Novelty of BlockDialect compared to existing block-wise mixed-precision quantization approaches**
> * Thanks for providing relevant references. We have carefully reviewed them to clarify the novelty of BlockDialect. While block-wise mixed-precision (MP) quantization methods and our mixed-format (same precision) approach share similarities, they differ in key aspects:
>   * MP requires either multiple types of MAC units or a complex modularized MAC with different precision modes, whereas our approach operates efficiently with a single low-precision MAC unit, improving hardware efficiency.
>   * MP introduces irregular memory access patterns or requires reordering mechanisms to manage distributed precision blocks, whereas our method maintains regular memory access without added complexity.
>   * MP relies on complex online tracking, pre-training or calibration with sample datasets to determine precision allocation before quantization, which may limit scalability or adaptability, whereas our approach avoids these requirements entirely.
>   * Due to these factors, MP demands extensive modifications to computation paths and kernel code, while our method only requires integrating quantization/dequantization functions before or after matrix multiplication, making it more portable and easier to deploy.
> * Additionally, BlockDialect introduces several novel aspects compared to existing quantization techniques:
>   * **Sample dataset agnostic**: Many activation quantization methods rely on sample dataset calibration or pre-training to avoid the high overhead of online processing, sacrificing adaptability. In contrast, BlockDialect eliminates the need for calibration or training by employing an efficient two-stage format selection (the efficiency of the two-stage approach is analyzed in the response to Reviewer 2 (ID: 6poj), section "Resource overhead of real-time MSE calculation") and logical operation-based quantization, enabling practical online processing.
>   * **Handling unstructured outliers**: Unlike many activation quantization methods that rely on easily tracked or calibrated structured outliers (e.g., channel-wise magnitude mean), BlockDialect can also handle unstructured outliers through fine-grained block-wise outlier localization.
>   * **Awareness of fine-grained block distribution**: To the best of our knowledge, this is the first mixed-format approach that constructs a diverse set of format candidates based on profiling fine-grained block distributions, rather than selecting from a few predefined standard formats or modifying exponent bias.
>   * **Addressing MX format limitations**: We identify and mitigate the limitations of the MX format, a block-wise quantization format already adopted in commercial products.
>   * **Efficient INT MAC utilization**: By carefully selecting a 0.5 granularity for representable values, BlockDialect enables floating-point representation using scaled integers ($0.5\cdot integer$), allowing direct computation with efficient low-precision integer MAC units. Note that we do not require floating-point operations even for the scaling factor, as we use a power-of-two scaling factor, which can be handled by addition and shifting logic while preserving accuracy.

---

> > ### Comment · Reviewer_rDoa · 2025-04-07
> >
> > Thank you for the detailed rebuttal. I appreciate the authors' clarifications and additional results.
> >
> > I still encourage more systematic evaluation across diverse model families (e.g., GPT, hybrid architectures) to better establish the robustness and adaptability of BlockDialect.
> >
> > Overall, the authors' response improves my understanding of the method’s strengths and novelty, but I believe further validation on real-world deployment aspects is necessary for stronger confidence.
> >
> > I maintain my overall score as Weak Accept.

---

> > > ### Author Response · Authors · 2025-04-08
> > >
> > > * Thank you for your additional comments. To address your concerns, we extended our evaluation to include the GPT model and observed that BlockDialect consistently outperforms other data formats across most cases, consistent with our results on LLaMA, Mistral, OPT, and Phi architectures. While time constraints limited us to evaluating only the GPT model, this additional experiment further supports our claim that BlockDialect's block-wise format assignment - being independent of architecture-specific characteristics - does not rely on assumptions about structural outlier patterns or computation flow. As a result, we expect it to extend naturally to hybrid transformer architectures, which often interleave heterogeneous layer types. We believe this strengthens the case for BlockDialect’s broad applicability.
> > >
> > > ###### MMLU is omitted due to low scores; perplexity is measured with 1024-token sequences
> > > ||GPT2-1.5B||`Linear`|`Full`|
> > > |:-:|:-:|:-:|:-:|:-:|
> > > |**Format**|**BlkSize**|**Eff.bit.**|PL↓\|CR↑\|GL↑|PL↓\|CR↑\|GL↑|
> > > |FP16|-||17.4\|53.2\|48.7|17.4\|53.2\|48.7|
> > > |NVFP4|16|4.5|18.6\|50.9\|47.7|18.8\|50.4\|47.8
> > > ||32|4.25|18.5\|50.4\|47.2|18.8\|49.8\|47.2
> > > |MXFP4|16|4.31|19.0\|51.3\|48.3|20.3\|49.2\|47.7
> > > ||32|4.16|19.0\|51.3\|48.0|20.1\|50.0\|47.5
> > > |BDFP4|16|4.56|17.9\|51.8\|48.1|18.0\|51.4\|47.6|
> > > ||32|4.28|18.1\|51.9\|47.3|18.4\|51.1\|46.8|

---

### Official Review · Reviewer_6poj · 2025-03-13

**Overall Recommendation:** 3

**Summary:**

They proposed BlockDialect, a block-wise finegrained mixed format technique that assigns a per-block optimal number format from a formatbook for better data representation. DialectFP4 ensures energy efficiency by selecting representable values as scaled integers compatible with low-precision integer arithmetic.
1. They introduce DialectFP4, a formatbook of FP4 variants (akin to dialects) that adapt to diverse data distributions. Three core principles of formatbook: 1) minimizing wasted or underestimated ranges, 2) prioritizing the representation of larger magnitudes, and 3) ensuring hardware efficiency.
2. They propose a two-stage approach for online DialectFP4 activation quantization.

**Claims And Evidence:**

The claims made in the submission supported by clear and convincing evidence.

**Essential References Not Discussed:**

N/A.

**Experimental Designs Or Analyses:**

Experimental soundness.

**Methods And Evaluation Criteria:**

The methods make sense.

**Other Comments Or Suggestions:**

I suggest moving Figure 2 to page 5 for easy viewing. In addition, Figure 2 lacks an explanation for some belonging, such as "sink".

**Other Strengths And Weaknesses:**

The experimental design of this paper considers the latest methods (Quarot) and model series (LLaMA3-8B), which are comprehensive. The authors deserve credit for providing the code.

**Questions For Authors:**

1. The proposed method eliminates real-time MSE calculation for activation quantization and claims that MSE overlooks the magnitude of data elements. The authors should provide a comparison of resource overhead.
2. The experiments are conducted on models with up to 8B parameters. How does BlockDialect scale with even larger models, such as those with 70B+ parameters? Are there any additional challenges or optimizations needed to maintain its efficiency and accuracy in such scenarios?
3. The paper explores various block sizes and their impact on performance. However, does the optimal block size vary across different layers? Could a dynamic block size strategy be more effective, where different layers use different block sizes based on their specific characteristics?
4. Given the growing interest in hybrid quantization techniques, how does BlockDialect interact with other methods? Please discuss the possibility of further combining with other strategies like SmoothQuant.
5. While the paper focuses on accuracy and energy efficiency, real-time inference latency is also crucial for practical deployment. How does BlockDialect impact the inference latency compared to full-precision models and other quantization methods? Are there any specific hardware accelerators that could further optimize the latency of BlockDialect?
6. The evaluation focuses on common-sense reasoning tasks. How does BlockDialect affect performance on other downstream tasks, such as machine translation and text summarization? Are there any specific tasks where BlockDialect might show more pronounced benefits or limitations?

**Relation To Broader Scientific Literature:**

Efficiency, model compression, and quantization.

**Theoretical Claims:**

Probably correct.

---

> ### Author Rebuttal · Authors · 2025-03-31
>
> **Resource overhead of real-time MSE calculation**
> * Qualitatively, MSE-based method requires 16 rounds (per dialect) of quantization, each involving FP16 square mean error accumulations for every block element, whereas our 2-stage selection efficiently operates in a single pass using 5-bit fixed-point values, logical operations, and simple counting.
> * Quantitatively, we design in SystemVerilog and synthesize quantization logic implementations with a 130nm process. For a fair comparison, we aim to match the latencies as closely as possible while evaluating area and power. Additionally, since FP16 operations in the exact MSE-based approach incur significant overhead, we convert into fixed-point representation and truncate the lower bits to reduce the complexity. Nevertheless, as shown in the table, MSE-based logic is 9.3× larger and consumes 10× more power. Notably, MSE-based logic fails to meet timing at 100MHz (synthesized at 83.3MHz), whereas our logic meets timing constraints even at 250MHz, underscoring the efficiency of our approach and the impracticality of online MSE-based selection.
>
> |Format selection method|Synthesized freq. ($MHz$)|Latency ($clock\ cycle$)|Power ($mW$)|Area ($\mu m^2$)|
> |:-:|:-:|:-:|:-:|:-:|
> |2-stage based|100|5|0.7|42833.6|
> |MSE-based|83.3|8|6.9|399409.3|
>
> **Scalability of BlockDialect**
> * Model size upper bound is due to our GPU resource constraints, not BlockDialect limitations. A key advantage of block-wise quantization is its independent block processing, which ensures efficiency and accuracy remain consistent regardless of model size. While there is a potential concern regarding the linear increase in per-block metadata as the model scales, our results in the paper shows that we can keep this overhead below 0.3 bits per data. Additionally, as discussed in the following response, dynamic block sizing (e.g., larger blocks for quantization-insensitive layers) can help mitigate this overhead.
>
> **Dynamic block size strategy**
> * That's a great question. In Table 3 (page 8), we analyze the impact of the dynamic block size strategy and find that the sensitivity of each sub-layer varies. Additionally, using larger blocks for most layers, while reserving smaller blocks for quantization-sensitive sub-layers, improves accuracy in certain cases. Based on this observation, we expect that selectively applying small block to quantization-sensitive layers (e.g., those with more outliers or greater influence on the final output) could enhance overall performance.
>
> **Combining with other strategies**
> * In Appendix G, we present the experiment results and discussion on combining BlockDialect with SmoothQuant. In summary, the combination leads to overall improvement, but with limited gains, indicating that the two methods are not entirely orthogonal. A more refined approach, such as applying smoothing exclusively to 'extreme' outliers before applying BlockDialect, may be beneficial. Additionally, we further explored combining BlockDialect with the Hadamard transformation-based rotation method and observed similar trends (limited gains) as in the BlockDialect-SmoothQuant hybrid quantization.
>
> **Impact on the inference latency**
> * BlockDialect offers clear latency benefits: (1) higher throughput with full-path low-precision MACs, unlike methods relying on FP16 MACs after dequantization or some matrix multiplications, and (2) reduced data movement latency from 4-bit weight/activation quantization (including KV cache), both crucial for inference latency. To ensure these gains are not offset by additional logic overhead, we assess this in our response to Reviewer 4 (ID: 8c5N) (see the “overhead of quantization/dequantization” section). While BlockDialect is deployable on GPUs, further optimizations, such as specialized MACs for our formatbook and efficient quantization/dequantization logic, will reduce latency. To achieve this, we are also taping out a hardware accelerator to incorporate these optimizations.
>
> **Workload sensitivity of BlockDialect**
> * We conducted additional experiments with different networks and downstream tasks (GLUE benchmark). Please see our response to Reviewer 4 (ID: 8c5N), section "Performance comparison of BlockDialect, NVFP4, and MXFP4 across various model sizes and architectures". BlockDialect uses a power-of-two shared exponent and smaller block sizes compared to conventional scale factor methods, preserving accuracy through optimal dialect assignment. This allows BlockDialect to effectively localize outliers, making it particularly beneficial for tasks with high magnitude variation and unstructured outliers.
>
> **Sink in Figure 2**
> * Thanks for your valuable feedback. We will reorganize the layout and clarify any ambiguities in the next revision. If the term "sink" refers to the range between 0 and 4 in the Figure 2(b), this occurs because we normalize the values using $2^{MaxExponent-2}$ , making the maximum magnitude become $2^2\cdot mantissa\ ([1,2))\rightarrow [4,8)$.

---

### Official Review · Reviewer_Vqff · 2025-03-14

**Overall Recommendation:** 4

**Summary:**

The paper introduces a block-wise finegrained mixed format technique (called BlockDialect) that assigns an optimal number format to each block and FP4 variants data format (called DialectFP4) that is built on shared exponent among a group of numbers. They also propose the method of efficient online quantization/dequantization and DialectFP4 computation method. They demonstrate that the proposed approach outperforms existing methods across multiple LLMs while leveraging lowprecision, energy-efficient MAC unit.

**Claims And Evidence:**

Some claims lack sufficient support. For instance:
1. The author states that "We introduce BlockDialect, a novel block-wise finegrained mixed format technique that assigns an optimal number format to each block" However, the proposed 4-bit BlockDialect may not be the optimal format; alternative methods, such as codebook-based quantization, could theoretically offer better performance.
2. Additionally, the author mentions in Section 3.1 that subtracting 2 from the shared exponent allows for a direct comparison with FP4 E2M1, but there is no explanation or compelling evidence provided to support this claim.

**Essential References Not Discussed:**

1. Hu X, Cheng Y, Yang D, et al. "I-LLM: Efficient Integer-Only Inference for Fully Quantized Low-Bit Large Language Models." arXiv preprint arXiv:2405.17849, 2024. (This paper mainly focuses on how to quantize all activations and weights in LLMs to enable fully integer-based computations in hardware, which is a good exploit of software-hardware co-design.)
2. Yuan Z, Shang Y, Zhou Y, et al. Llm inference unveiled: Survey and roofline model insights[J]. arXiv preprint arXiv:2402.16363, 2024. (This paper provides an overview of how different methods impact inference efficiency from the perspective of software-hardware co-design.)

**Experimental Designs Or Analyses:**

I carefully reviewed the author's experimental design.

### Model and Dataset Selection
The author utilized the LLaMA-2-7B, LLaMA-3-8B, and Mistral-7B models, evaluating them on the WikiText2 dataset while employing zero-shot commonsense reasoning tasks .
Strengths: The selection of widely used LLM models and standard datasets ensures the generalizability and comparability of the experiments.
Potential Issues: All chosen models are around 7 billion parameters in size, without consideration for networks of varying sizes. Besides, the MMLU is not used.

### Dataformat Compare
The study compares BlockDialect with baseline methods such as MXFP4, LLM-FP4, and Quarot. A variety of quantization methods were selected as baselines, encompassing both hardware-supported and software-supported quantization techniques, ensuring a comprehensive comparison.
Potential Issues: The comparison does not include codebook-based methods, such as Vector Quantization. I wonder whether table-based approaches could potentially achieve similar or even higher performance through search methods, although storing the table entries might result in slightly increased storage requirements.

### Effective Bit Width Calculation
The experimental design calculated the effective bit width (Eff. bit), taking into account the overhead of scaling factors or dialect identifiers. By calculating the effective bit width, the design more accurately reflects the memory footprint and computational efficiency of the quantization methods.
Potential Issues: The specific calculation method for effective bit width is not detailed, which may lead to a lack of transparency in the results.

### Evaluation of Hardware Implementation
The author modeled MAC units at different precision levels using SystemVerilog and synthesized them with Synopsys Design Compiler to evaluate area and power consumption. The assessment of hardware implementation demonstrates the efficiency of BlockDialect in practical hardware contexts.
Potential Issues: The specific parameters for the hardware evaluation (e.g., clock frequency, process node) are not clearly stated, which could affect the comparability of the results. Additionally, the reported findings indicate that the method proposed by the author shows significantly lower overhead for the multiplier compared to INT5, raising questions about the validity of the evaluation settings.

**Methods And Evaluation Criteria:**

The proposed methods make sense for energy-efficient LLM inference.
The evaluation criteria also makes sense.

**Other Comments Or Suggestions:**

The comparison does not include codebook-based methods, such as Vector Quantization. I wonder whether table-based approaches could potentially achieve similar or even higher performance through search methods, although storing the table entries might result in slightly increased storage requirements.

**Other Strengths And Weaknesses:**

### Strengths
1. Unlike existing methods that primarily focus on "how to scale," this paper introduces a new perspective of "how to represent" each block, removing the reliance on a single scaling strategy and thereby better capturing the data distribution within the block.
2. The hardware cost is considered in the design of data format, which is a successful explore of software-hardware co-design for efficient LLM inference.
3. Experimental results show that BlockDialect significantly outperforms the existing MXFP4 format across multiple LLM models, especially in full-path matrix multiplication quantization, with only a 5.45% (LLaMA3-8B) and 2.69% (LLaMA2-7B) accuracy drop compared to full precision, while reducing the bit usage per data.


### Weaknesses
1. One of the main drawback of this paper lies in the insufficient clarity of the methodology section. Without referring to the code, the entire methodology section is difficult to fully understand.
2. There are some questions when I execute the code (See Question for authors).

**Questions For Authors:**

1. Subtracting two facilitates direct comparison with FP4 E2M1 ? Why? Can you provide any proof?
2. Hardcoded DialectFP4—where do these numbers come from? Why were these specific values chosen, such as 4.5, 7.5, 5.5, and 5.0, which cannot be represented by FP4-E2M1? What is the purpose of selecting these? I find the author's description very difficult to understand. Moreover, when I ran the W4A4Linear using the author's code, I found numbers like 0.25 and 0.275, which FP4 cannot represent at all. Why is that?
3. The author mentions in Figure 5 the operation man << exp, followed by truncation, introducing an intermediate 5-bit data type. Why is this 5-bit data type used instead of other intermediate data types? How was the choice of 5 bits made? Is there proof that this is a result of a trade-off between precision and bit count?
4. When sharing the exponent, why use log2 to determine the exponent of the absolute maximum value within a block, when bit manipulation could be employed directly (since the data format is based on FP16)?

**Relation To Broader Scientific Literature:**

### Block-wise Quantization
Block-wise quantization is a widely adopted technique that assigns scaling factors on a per-block basis to constrain the impact of outliers.
The author use block-wise quantization is the same as previous methods.

### Non-Uniform Quantization

Non-uniform quantization serves as an alternative to integer formats, aiming to better capture data distributions in large language models. Floating-point formats excel in handling the wide value ranges encountered in deep learning models, while lookup-based formats better align with the distributions of large language models through statistical distribution quantile functions.

This paper introduces DialectFP4, a set of FP4 variants (akin to "dialects") tailored for diverse block-level data distributions, and achieves online DialectFP4 activation quantization through a practical two-stage approach. Compared to existing non-uniform quantization methods, this approach offers greater flexibility and hardware efficiency.

### Activation Quantization

Activation quantization faces challenges such as real-time execution, large dynamic ranges, and inter-channel outliers. Existing methods include mixed-precision subgrouping, migrating quantization difficulty to weights, and using Hadamard matrices to reduce outliers.

This paper achieves efficient activation quantization by introducing FP4 variants and adopting a two-stage approach for online optimal format selection. Compared to existing activation quantization methods, this approach reduces reliance on high-precision operations, improving energy efficiency and inference speed.

**Theoretical Claims:**

There is no theoretical claims in this paper.

---

> ### Author Rebuttal · Authors · 2025-03-31
>
> **Reason for Subtracting 2 from the Shared Exponent**
> * When normalizing by the maximum exponent, normalized values fall within [0, 2), whereas FP4 variants span [0, 7.5]. Subtracting 2 from the shared exponent extends the range to [0, 8), enabling FP4 variants to represent normalized values without additional scaling. This also enables direct comparisons and highlights FP4 E2M1's limitations, such as its inability to represent values in the (6, 8) range within a power-of-two dynamic range.
>
> **How to determine the representable values of DialectFP4?**
> * The representable values in DialectFP4 are designed to capture diverse block distributions that a single FP4 E2M1 format cannot express.
>   * To account for varying dynamic ranges of fine-grained blocks, each dialect's largest value is selected from a range of 7.5 to 4.0, while second-largest values vary to adapt to large-magnitude distributions.
>   * In the second stage of format selection, we choose the best dialect based on how many block values fall within each dialect's beneficial range. We adjust the second-largest value, which only varies between dialects with the same maximum, to ensure the widths of the beneficial ranges of the compared dialects are similar. This approach helps avoid bias toward any particular dialect.
>   * The representable values follow a fixed granularity of 0.5, ensuring all values are scaled integers (integer x 0.5). This facilitates efficient integer MAC operations. Additionally, six values remain consistent across dialects to simplify quantization/dequantization logic.
> * While DialectFP4 demonstrates strong performance, we acknowledge that it is not optimally constructed. There is potential to adaptively generate DialectFP4 tailored to specific models or layers.
> * Unexpected numbers may result from the shared exponent applied after quantization. For example, with a shared exponent of -3, a quantized value of 3.0 reverts to 0.375 (3/8). If this doesn't apply, please clarify where your values appear.
>
> **Justification for the 5-bit intermediate representation**
> * The representable magnitudes of DialectFP4 range from 0.0 to 7.5 with a 0.5 granularity, requiring three integer bits and one fractional bit. However, to handle rounding accurately, an additional fractional bit is needed to determine whether to round up or down to the nearest representable value, resulting in a total of 5 bits.
>
> **We appreciate the feedback and will clarify the points explained above in the next revision.**
>
> ---
> **Limited network and workload diversity**
> * We conduct experiments across networks of varying sizes and additional workloads, such as MMLU. For further details, please refer to our response to Reviewer 4 (ID: 8c5N), “Performance comparison of BlockDialect, NVFP4, and MXFP4 across model sizes and architectures” section.
>
> **Comparison with table-based approaches and I-LLM**
> * While table-based approaches with high-precision entries may perform well, they typically rely on high-precision MACs, which conflicts with our goal of enabling low-precision MACs for better hardware efficiency. Additionally, codebook-based methods are generally restricted to weight-only quantization due to the challenges of adaptive codebook construction and online activation codebook quantization. This is why we exclude this method in our weight-activation quantization comparison.
> * We appreciate the mention of relevant papers. I-LLM functions as an adaptive online variant of SmoothQuant, migrating quantization difficulty to easier-to-quantize matrices, albeit with additional tracking overhead. As discussed in Appendix G, we believe such approaches can complement BlockDialect by managing outliers at different levels. In this context, I-LLM serves as a valuable reference for improving accelerators using BlockDialect.
>
> **Effective bitwidth calculation**
> * Due to character limitations, we have integrated this response with our reply to Reviewer 3 (ID: rDoa) under the "Metadata Storage Requirements" section.
>
> **Hardware evaluation parameters**
> * As stated in the implementation section and the caption of Table 5, we use a 0.5 GHz clock speed and a 45 nm process node for our hardware evaluation.
>
> **Comparison between proposed MAC and INT5 MAC**
> * The discrepancy mainly arises from the sign-magnitude (Ours) vs. 2's complement (INT5) implementation. As recent works$^{[1,2]}$ have noted, sign-magnitude representation offers advantages over 2's complement MAC due to its much lower toggle rate and simpler sign processing.
>   - ###### [1] Wang et al. "14.3 A 28nm 17.83-to-62.84 TFLOPS/W Broadcast-Alignment Floating-Point CIM Macro with Non-Two's-Complement MAC for CNNs and Transformers." ISSCC, 2025. [2] Han & Chandrakasan. "MEGA. mini: A Universal Generative AI Processor with a New Big/Little Core Architecture for NPU." ISSCC, 2025.
>
> **Use of log2 function**
> * That’s a great point. We just used log2 as it seems intuitive, but will consider changing using an efficient bit manipulation.

---

### Decision · Program_Chairs · 2025-05-01

**Decision:**

Accept (poster)

**Comment:**

This paper proposes BlockDialect, a block-wise, fine-grained mixed-format techniques that assigns an optimal number format to each block from a predefined format book, aiming to improve the FP4 format for low-precision inference of large language model (LLMs). The method is validated on popular LLMs, demonstrating improved performance in both accuracy and hardware efficiency.

The paper is well written, and the algorithm design is clearly presented. Unlike traditional quantization methods that rely on per-block scaling or mixed precisions, this paper assigns a specific number format to each block to better present block-wise data distributions, which is a novel approach. The proposed efficient two-stage format selection process, as well as the insights into block-level data distribution, are valuable contributions to the research in low-precision quantization of LLMs.

During rebuttal, the authors provide additional analysis on quantization/dequantization overhead and a comparison with latest NvFP4 format, further strengthening the paper. Therefore, this paper is recommended for acceptance.